## RESEARCH ARTICLE

# PIKfyve is required for efficient phagosomal Rab7 acquisition and the delivery and fusion of early macropinosomes to phagosomes

James H. Vines, Catherine M. Buckley, Ilona Willson, Daniel S. Stark and Jason S. King*

## ABSTRACT

Phagosome maturation is tightly regulated to ensure efficient delivery of the complex arsenal of antimicrobial activities that kill and digest captured microbes. Like other endocytic pathways, phagosome maturation is regulated by a combination of Rab GTPases and phosphoinositide (PIP) signalling lipids, which define membrane identity and recruit specific effectors. PIKfyve is a phosphoinositide 5-kinase, which converts phosphatidylinositol 3-phosphate [PI(3)P] into phosphatidylinositol 3,5-bisphosphate [PI(3,5)$P_2$] on endosomes. Disruption of PIKfyve results in severe defects in phagosomal maturation but the underlying mechanism remains unclear. Here, we use the model professional phagocyte, *Dictyostelium discoideum* to dissect the role of PIKfyve in the crucial first steps of phagosome maturation. We find that, although early Rab5 dynamics are unaffected, loss of PIKfyve reduces phagosomal Rab7 acquisition by preventing fusion with a pool of Rab7 and V-ATPase positive endosomes. By following PIP dynamics using our recently characterised PI(3,5)$P_2$ probe SnxA, we delineate multiple subpopulations of Rab7-positive endosomes that fuse sequentially with phagosomes. We identify one of these as PI(3,5)$P_2$-positive macropinosomes, which dock and fuse with phagosomes in a PIKfyve-dependent manner. We therefore show that *Dictyostelium* phagosomes accumulate Rab7 by PIKfyve-dependent vesicular fusion in addition to acquisition from a cytosolic pool. We identify PI(3,5)$P_2$-positive macropinosomes as a specific subset of fusogenic vesicles, which we propose enables content mixing and the efficient bulk delivery of lysosomal components to phagosomes.

KEY WORDS: Phagocytosis, Macropinocytosis, PIKfyve, Rab7, *Dictyostelium*

## INTRODUCTION

The degradation of endocytic cargo is a highly regulated cellular process. Endosomes must acquire the correct markers with temporal accuracy to facilitate their maturation. This is achieved through sequential regulatory steps, which identify compartments in different stages of maturation. In this way, effector proteins and lysosomal compartments are recruited over the course of maturation to mediate the recycling or degradation of internalised material.

School of Biosciences, University of Sheffield, Firth Court Western Bank, Sheffield S10 2TN, UK.

*Author for correspondence (Jason.King@sheffield.ac.uk)

C.M.B., 0000-0003-2105-0456; J.S.K., 0000-0003-0596-4506

Disruption of any one of these key regulators can result in aberrant endosomal trafficking and severe physiological defects.

Inositol phospholipids define specific compartment membranes during endosomal maturation. They contain a glycerol backbone, two non-polar fatty acid tails, and an inositol head group, which can be phosphorylated at any of three positions (3, 4 or 5) to form one of eight different phosphoinositide (PIP) species. Each PIP can recruit different effector proteins, allowing for spatially segregated protein enrichment to different membranes. Interconversion of PIP species by the activity of a large family of phosphatases and kinases therefore allows specific effectors to be recruited at specific times throughout maturation (Bohdanowicz and Grinstein, 2013).

PIKfyve is a phosphoinositide 5-kinase that is recruited to early endosomes or the yeast vacuole via a FYVE (Fab1, YOTB, Vac1 and EEA1) domain, which specifically binds to phosphatidylinositol 3-phosphate [PI(3)P] (Cabezas et al., 2006; Sbrissa et al., 1999; Yamamoto et al., 1995). There, it phosphorylates PI(3)P to produce phosphatidylinositol 3,5-bisphosphate [PI(3,5)$P_2$], one of the least abundant and well-understood PIPs. Across an array of organisms, disruption of PIKfyve causes a range of trafficking defects, including characteristic swollen endosomes and defects in lysosomal degradation (Buckley et al., 2019; Choy et al., 2018; de Lartigue et al., 2009; Dove et al., 2009; Ikonomov et al., 2001; Kim et al., 2014; Krishna et al., 2016; Nicot et al., 2006). Precisely how PIKfyve disruption leads to these phenotypes is unclear, although several PI(3,5)$P_2$-activated ion channels on endosomal membranes have been identified, including TRPML1 and TPC2, which have been implicated in endosome–endosome fusion events (Dong et al., 2010; Leray et al., 2022; Samie et al., 2013; Wang et al., 2012).

The Rab family of small GTPases also play key roles in regulating endolysosomal traffic (Borchers et al., 2021). Early endosomes are marked by Rab5 family proteins (Rab5A and Rab5B in *Dictyostelium*, and Rab5a, Rab5b and Rab5c in mammals, hereafter generically referred to as Rab5), which recruits effector proteins essential for early endosome fusion events (Henry et al., 2004; Lippuner et al., 2009), including Vps34, the class III phosphoinositide 3-kinase (PI3K) responsible for PI(3)P synthesis. As maturation progresses, Rab5 is exchanged with the lysosomal marker, Rab7 family proteins (Rab7A and Rab7B in *Dictyostelium*, and Rab7a and Rab7b in mammals, hereafter generically referred to as Rab7). This switch has been shown to be mediated by the evolutionarily conserved Mon1–Ccz1 complex, which coordinates the dissociation of Rab5 with recruitment and activation of Rab7 on endosomes (Kinchen and Ravichandran, 2010; Langemeyer et al., 2020; Nordmann et al., 2010; Poteryaev et al., 2010). Accumulation of Rab7 on late endosomes promotes fusion with lysosomal compartments and its disruption results in severely perturbed lysosomal delivery to phagosomes (Rupper et al., 2001).

Our previous work has shown that PIKfyve is crucial for phagosome maturation in the soil-dwelling amoeba *Dictyostelium discoideum* (Buckley et al., 2019). *Dictyostelium* is a well-characterised professional phagocyte, which feeds through

phagocytosis of bacteria or bulk uptake of media by the related process of macropinocytosis. These processes are well-conserved across evolution (Boulais et al., 2010; King and Kay, 2019), and the amenability of *Dictyostelium* to genetic manipulation, biochemical analysis and fluorescence microscopy studies make it a useful model system.

*Dictyostelium* mutants lacking PIKfyve (ΔPIKfyve) have severe defects in phagosomal proteolysis and killing (Buckley et al., 2019). These defects are, in part, caused by perturbed V-ATPase and hydrolase delivery to newly formed phagosomes. Importantly, loss of PIKfyve renders *Dictyostelium* hypersensitive to infection with *Legionella pneumophila*, demonstrating the physiological importance of PIKfyve in innate immunity. PIKfyve is also important for phagosome–lysosome fusion in both macrophages and neutrophils (Dayam et al., 2015, 2017; Isobe et al., 2019; Kim et al., 2014). This indicates a central and conserved role, but how PIKfyve activity is regulated and integrates with other components of the phagosomal maturation machinery, such as Rab signalling, and fusion with other endosomal compartments is poorly understood.

Here, we dissect how PIKfyve disruption causes defective phagosome maturation. By examining key maturation markers [Rab5, PI(3)P, V-ATPase and Rab7], we show that although PIKfyve-deficient phagosomes acquire and recycle Rab5 normally, they accumulate almost no Rab7, which appears to be normally delivered by vesicle fusion. Using our recently described reporter for PI(3,5)P$_2$ (SnxA) (Vines et al., 2023), we identify an additional route for Rab7 enrichment via fusion with PI(3,5)P$_2$-positive macropinosomes. We therefore show PIKfyve has a specific role in regulating heterotypic fusion between phagosomes and a temporally defined population of macropinosomes, which we suggest plays a previously unappreciated role in maintaining digestive efficiency.

## RESULTS

### PIKfyve is required for Rab7 delivery to phagosomes

To understand the functional role of PIKfyve, we examined how its loss affected other core components of the phagosome maturation pathway. Previously, we have observed that PIKfyve–GFP is transiently recruited to *Dictyostelium* phagosomes between 30 s and 120 s following engulfment (Vines et al., 2023), so we focused our analysis on components likely to be active within this time frame. During classical endocytosis, Rab5 accumulates at early stages and exchanges with Rab7 as endosomes mature (Rink et al., 2005). A similar Rab5-to-Rab7 exchange has been described during both phagosome and macropinosome maturation around the same time we observe PIKfyve–GFP recruitment (Kerr et al., 2006; Poteryaev et al., 2010; Tu et al., 2022; Vieira et al., 2003).

Co-expression of RFP–Rab5A and GFP–Rab7A allowed us to follow the dynamics of both proteins to newly formed phagosomes in wild-type and ΔPIKfyve cells (Fig. 1A,B; Movie 1). In wild-type cells, as previously reported (Tu et al., 2022), RFP–Rab5A was highly enriched on perinuclear endosomes and to a lesser extent at the plasma membrane. RFP–Rab5A was further recruited to phagosomes

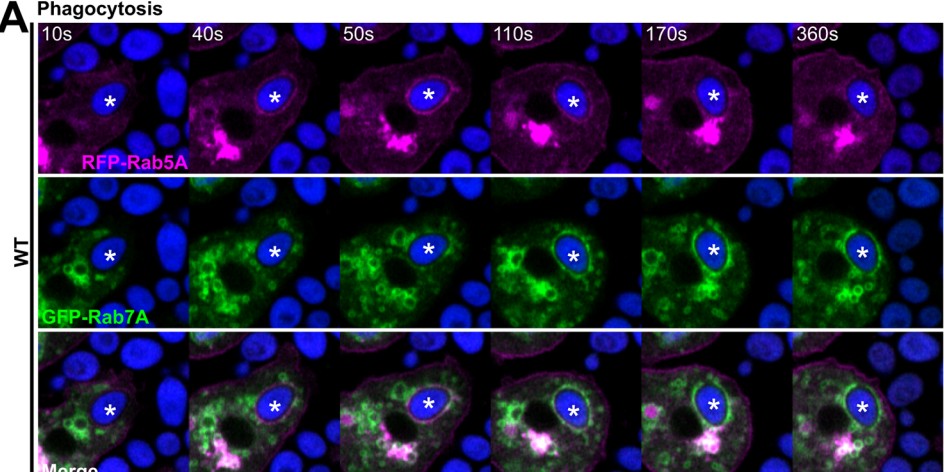

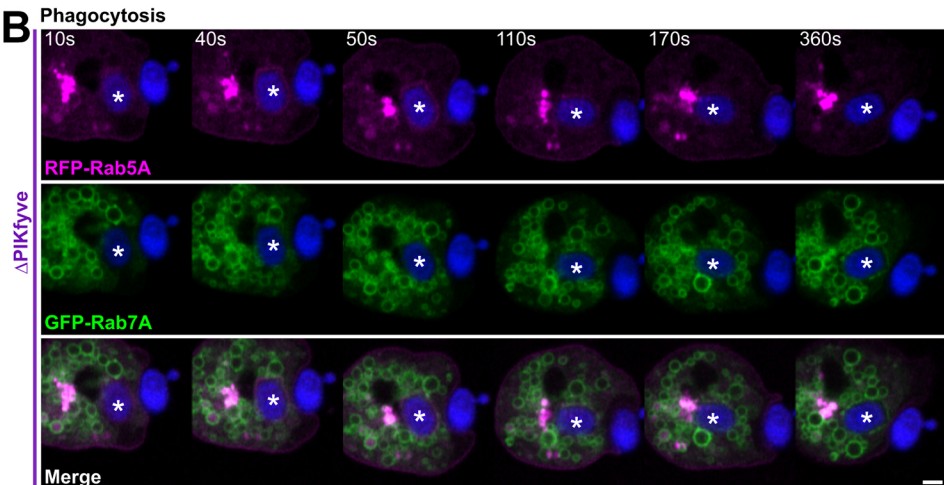

**Fig. 1. Simultaneous analysis of RFP–Rab5 and GFP–Rab7 recruitment.** Timelapse movies of (A) wild-type (WT) and (B) ΔPIKfyve cells co-expressing RFP–Rab5A (magenta) and GFP–Rab7A (green) during phagocytosis of Alexa Fluor 405-yeast (blue). Rab7A only arrives on wild-type phagosomes. The followed phagosome is indicated by an asterisk, and the bright magenta structure is the perinuclear Rab5 pool. The full timeseries is shown in Movie 1. Images representative of more than ten experimental repeats. Scale bar: 2 μm.

following internalisation before its removal in the following minutes. As expected, this coincided with a gradual accumulation of GFP–Rab7A, which also localized to a large pool of small endosomes that continuously clustered around phagosomes and appeared to fuse from ∼30 s post-engulfment. In contrast, although RFP–Rab5A dynamics appeared identical in ΔPIKfyve cells, phagosomes only ever accumulated very low levels of GFP–Rab7A, despite strong recruitment to other enlarged endocytic compartments (Fig. 1B).

To quantify changes in protein enrichment around engulfed particles over multiple events, we employed an automated image analysis pipeline, which we have previously described (Vines et al., 2023). This segments the yeast and measures the intensity of fluorescent fusion proteins around the periphery at each time point. To reduce the possibility of artefacts from overexpressing both proteins during quantification, Rab5A and Rab7A were expressed individually as GFP fusions. Multiple events were then averaged and normalised to the intensity at the moment engulfment completed (Fig. 2).

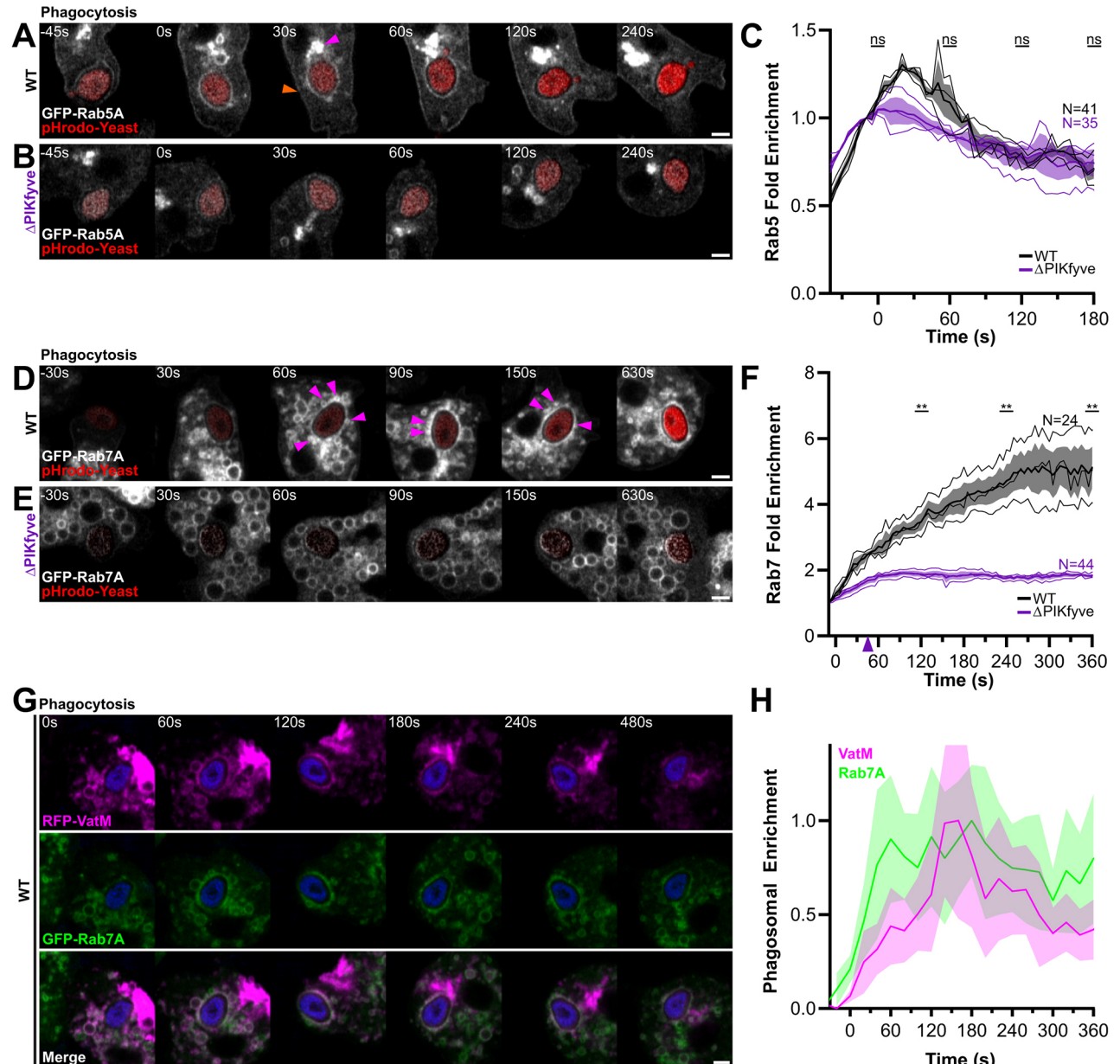

Fig. 2. Loss of PIKfyve affects Rab7 recruitment to phagosomes. (A,B) Timelapses of GFP–Rab5A during phagocytosis of pHrodo-yeast (red) in wild-type (WT; A) and ΔPIKfyve (B) cells. GFP–Rab5A is constitutively enriched on a perinuclear pool (pink arrowhead) and the plasma membrane (orange arrowhead) as well as the phagosomal envelope for 0–60 s. See Movie 2 for complete sequence. (C) Quantification of GFP–Rab5A enrichment averaged over the indicated number of independent events. No significant differences were found at any timepoint tested (paired two-tailed t-test at 60 s intervals). (D,E) GFP–Rab7A recruitment to phagosomes in wild-type and ΔPIKfyve cells, respectively. Arrowheads indicate the clustered GFP–Rab7A-enriched vesicles only observed in wild-type cells. See Movie 5 for complete sequence. (F) Quantification of phagosomal GFP–Rab7A enrichment, demonstrating a significant decrease in ΔPIKfyve mutants (paired two-tailed t-test at the time points indicated). (G) Timelapse movie of wild-type cells co-expressing RFP–VatM (magenta) and GFP–Rab7A (green). The large bright magenta structure is the contractile vacuole network. (H) Fluorescence intensity around the phagosome shown in G. Data is mean±s.d. intensity of a linescan along the phagosomal membrane normalised between minimum and maximum intensities observed. Data in G and H representative of five experimental repeats. Scale bars: 2 µm.

Journal of Cell Science

This analysis confirmed the enrichment of GFP–Rab5A starting just prior to phagocytic cup closure (Fig. 2A–C; Movie 2). This was independent of successful internalisation (Movie 3), suggesting a potential role for Rab5 before engulfment is complete in *Dictyostelium*. GFP–Rab5A then intensified following internalisation and peaked 30 s after engulfment (Fig. 2A). Although expression levels were lower in the mutants (Fig. S1A), the timing of GFP–Rab5A recruitment was not significantly changed in ΔPIKfyve cells (Fig. 2B,C).

As Rab5 is also responsible for PI(3)P synthesis, due to the recruitment and activation of the class III PI3K Vps34, we also quantified PI(3)P using the well characterised reporter GFP–2xFYVE (Gillooly et al., 2000). In wild-type cells, GFP–2xFYVE became enriched on phagosomes in a uniform ring within 60 s of engulfment (Fig. S1B–D; Movie 4) and was maintained for at least 50 min (Fig. S1E). PI(3)P also localized to a set of small vesicles throughout the cytosol (arrowheads in Fig. S1B). In ΔPIKfyve cells, however, these vesicles were replaced with large numbers of characteristically swollen endosomes, which moved slowly around the cell (Fig. S1C). Nonetheless, there were no quantifiable differences in phagosomal PI(3)P dynamics in ΔPIKfyve cells, consistent with normal Rab5 dynamics (Fig. S1D).

Quantification of GFP–Rab7A in wild-type cells showed a continuous accumulation to phagosomes starting at ~30 s post-engulfment and continuing for the next ~5 min (Fig. 2D–F; Movie 5). GFP–Rab7A was then maintained for at least 50 min (Fig. S1F,G), most likely until the transition to a post-lysosome (Buczynski et al., 1997). In ΔPIKfyve cells, however, quantitative analysis confirmed that the majority of phagosomal GFP–Rab7A was absent, although ~15% of the GFP–Rab7A signal did still accumulate within the first 60 s. This indicated that the majority of Rab7 accumulation is PIKfyve dependent, with a smaller fraction remaining in the absence of PIKfyve activity (Fig. 2F).

To test whether the defect in Rab7 accumulation is specific to the large yeast-containing phagosomes, we also studied GFP–Rab7A recruitment to phagosomes containing RFP-expressing *E. coli*. Although we were unable to reliably measure intensities over time in these smaller and more dynamic compartments, GFP–Rab7A strongly accumulated on almost every phagosome within 5 min of addition of bacteria in wild-type cells (Fig. S2). This was significantly delayed in ΔPIKfyve cells, although bacteria-containing phagosomes did still visibly accumulate GFP–Rab7A signal after 10 min. We cannot definitively conclude whether the amount of GFP–Rab7A on bacterial phagosomes is reduced at these later points compared to wild-type cells, but the slower accumulation is consistent with our previous data showing a similar delay, but not complete absence of bacterial killing in ΔPIKfyve cells (Buckley et al., 2019).

In these experiments, it was notable that although GFP–2xFYVE appeared as a smooth ring directly on the phagosomal membrane, in wild-type cells GFP–Rab7A-positive vesicles appeared to cluster around phagosomes before fusing. This was apparent by analysis of the variance in fluorescence along the phagosome membrane (Fig. S1H) and indicates that phagosomal Rab7 accumulation is at least partly due to fusion with other Rab7-containing compartments. In contrast, although the cytosol of ΔPIKfyve cells was full of swollen GFP–Rab7A-positive vesicles, there was no obvious clustering around the nascent phagosome (Fig. 2E).

We have previously shown that disruption of PIKfyve causes a reduction in both V-ATPase and lysosomal enzyme delivery to phagosomes (Buckley et al., 2019). We therefore asked whether Rab7 and V-ATPase were delivered on the same vesicles by co-expressing GFP–Rab7A with RFP–VatM (a component of the

V-ATPase complex) in wild-type cells (Fig. 2G). This showed that all VatM-positive vesicles were also Rab7A-positive, and both proteins accumulated on the phagosomal membrane concurrently (Fig. 2H). This indicates that PIKfyve is required for the delivery of Rab7A- and VatM-positive endosomes to the early phagosome and its absence causes a generalized loss in the delivery of the lysosomal components required for maturation.

## PIKfyve is not important for Rab7 delivery to macropinosomes

Although GFP–Rab7A did not accumulate on ΔPIKfyve phagosomes, it clearly still localised to other large endocytic compartments (Fig. 2E). As these resemble the swollen macropinosomes previously observed in these cells (Buckley et al., 2019), we asked whether the defect in GFP–Rab7A delivery was specific to phagosome maturation. As individual macropinosomes are difficult to follow for more than 2–3 min by timelapse microscopy, we labelled macropinosomes with a 2-min pulse of TexasRed–dextran before washing out into normal medium (Fig. 3A,B). Different fields of view were then captured at time intervals after dextran removal allowing the percentage of macropinosomes positive for GFP to be measured for up to 10 min (Buckley et al., 2016). In this assay, GFP–Rab7A recruitment to macropinosomes was indistinguishable between ΔPIKfyve cells and wild-type controls, accumulating within 2 min of internalisation (Fig. 3C).

Although GFP–Rab7A dynamics appeared normal, the macropinosomes in ΔPIKfyve cells still appeared much larger. To avoid any potential detrimental effects of overexpressing GFP–Rab7A, we therefore quantified macropinosomes size in untransformed cells, following the same TexasRed–dextran pulse-chase. Quantification of macropinosome size and fluorescence intensity demonstrated that although they started off the same size, the normal shrinkage and concentration of their lumen was significantly impaired in PIKfyve mutants (Fig. 3D–F). Therefore, shrinkage and Rab7 accumulation are functionally separable, indicating multiple independent roles of PIKfyve.

To functionally assess macropinosome maturation, we also measured proteolysis of the reporter dye DQ-BSA over time. Although loss of PIKfyve almost completely blocks proteolysis of DQ-BSA when conjugated to beads (Buckley et al., 2019), degradation of the fluid-phase dye was unaffected (Fig. 3G). Therefore, despite generally comparable maturation pathways, PIKfyve plays a specific role in the fusion of lysosomes to phagosomes, but not macropinosomes.

## PI(3,5)P$_2$ is delivered by fusion of small Rab7-positive vesicles

The evidence above suggests that PIKfyve or production of PI(3,5)P$_2$ is required for lysosomal fusion with phagosomes. To better understand how Rab7 is recruited relative to PI(3,5)P$_2$, we co-expressed RFP–Rab7A with PIKfyve–GFP in ΔPIKfyve cells (Fig. 4A). As expected, PIKfyve–GFP expression restored Rab7A recruitment to phagosomes, and recruitment of PIKfyve–GFP overlapped with the start of Rab7 enrichment (Fig. 4B). As we previously showed, PIKfyve recruitment was only transient and was lost from the phagosomal membrane within 2 min (Vines et al., 2023). The sustained presence of its product PI(3,5)P$_2$ is therefore likely to drive continued Rab7 recruitment.

To observe PI(3,5)P$_2$ dynamics, we used our recently identified biosensor, SnxA, which specifically binds this lipid with high affinity (Vines et al., 2023). In *Dictyostelium*, SnxA–GFP was enriched on both phagosomes and macropinosomes at ~120 s after

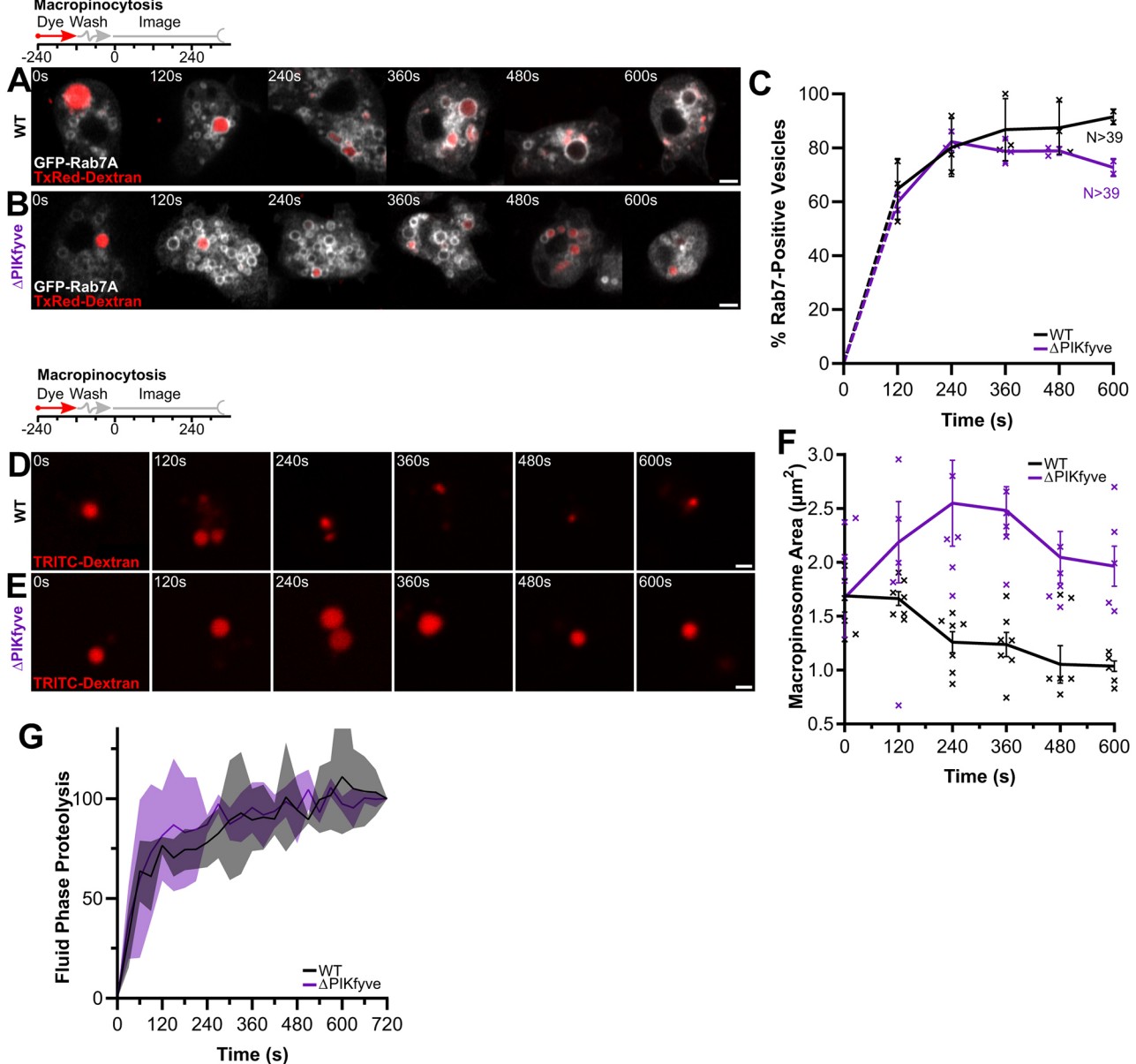

**Fig. 3. PIKfyve is not important for Rab7 delivery to macropinosomes.** (A,B) GFP–Rab7A recruitment to macropinosomes labelled by a 2-min pulse of TxRed–dextran (red) in wild-type (WT; A) and ΔPIKfyve(B) cells. (C) Quantification by scoring the percentage of GFP-positive macropinosomes at each time point after dextran wash out. GFP–Rab7A dynamics are similar in both cell types. Data shown is mean±s.e.m. of three independent experiments, with N indicating the total number of cells measured. Biological replicates shown by crosses. (D,E) Labelling of macropinosomes by a 2-min pulse of TRITC–dextran (red) in wild-type and ΔPIKfyve cells, respectively. (F) Quantification of macropinosome shrinkage during early maturation. Wild-type macropinosomes lose approximately half their volume within 5 min, but this is disrupted in ΔPIKfyve cells. Graph represents the mean±s.e.m. of six independent experiments. Biological replicates shown by crosses. All values for ΔPIKfyve cells after 120 s are statistically significant from control cells ($P<0.001$, two-way ANOVA with multiple comparisons). (G) Fluid phase proteolytic activity in wild-type and ΔPIKfyve cells measured by increase in fluorescence after a 2-min pulse of DQ-BSA and measurement of fluorescence on a plate reader. Data shown is mean±s.d. of three independent experiments. Scale bars: 2 μm.

engulfment, just before PIKfyve itself left the phagosome. Upon disruption of PIKfyve, SnxA–GFP became completely cytosolic. For clarity, the dynamics of $PI(3,5)P_2$ during phagosome maturation relative to other markers from this and other studies is summarised in Fig. 4C.

Closer examination of SnxA–GFP recruitment to phagosomes and macropinosomes by timelapse microscopy, indicated that accumulation of $PI(3,5)P_2$ was accompanied by the docking and fusion of multiple smaller SnxA–GFP-positive vesicles less than 1 μm in size. These cluster around macropinosomes and phagosomes several seconds before they acquire SnxA-GFP

themselves (arrowheads in Fig. 4D,E; Movie 6). This implies that $PI(3,5)P_2$ accumulates through both *in situ* synthesis by PIKfyve and delivery via fusion with another $PI(3,5)P_2$ compartment. This suggests a role for PIKfyve in regulating the docking or fusion of a specific subset of vesicles to the phagosome.

We then asked whether the small $PI(3,5)P_2$-positive vesicles also contained Rab7 by co-expressing SnxA–GFP with RFP-Rab7A. This demonstrated that all docked SnxA-positive vesicles were also positive for Rab7A (white arrowheads in Fig. 4F and Fig. S3A). However, RFP–Rab7A also localised to additional populations of vesicles that started to cluster and fuse much earlier than those

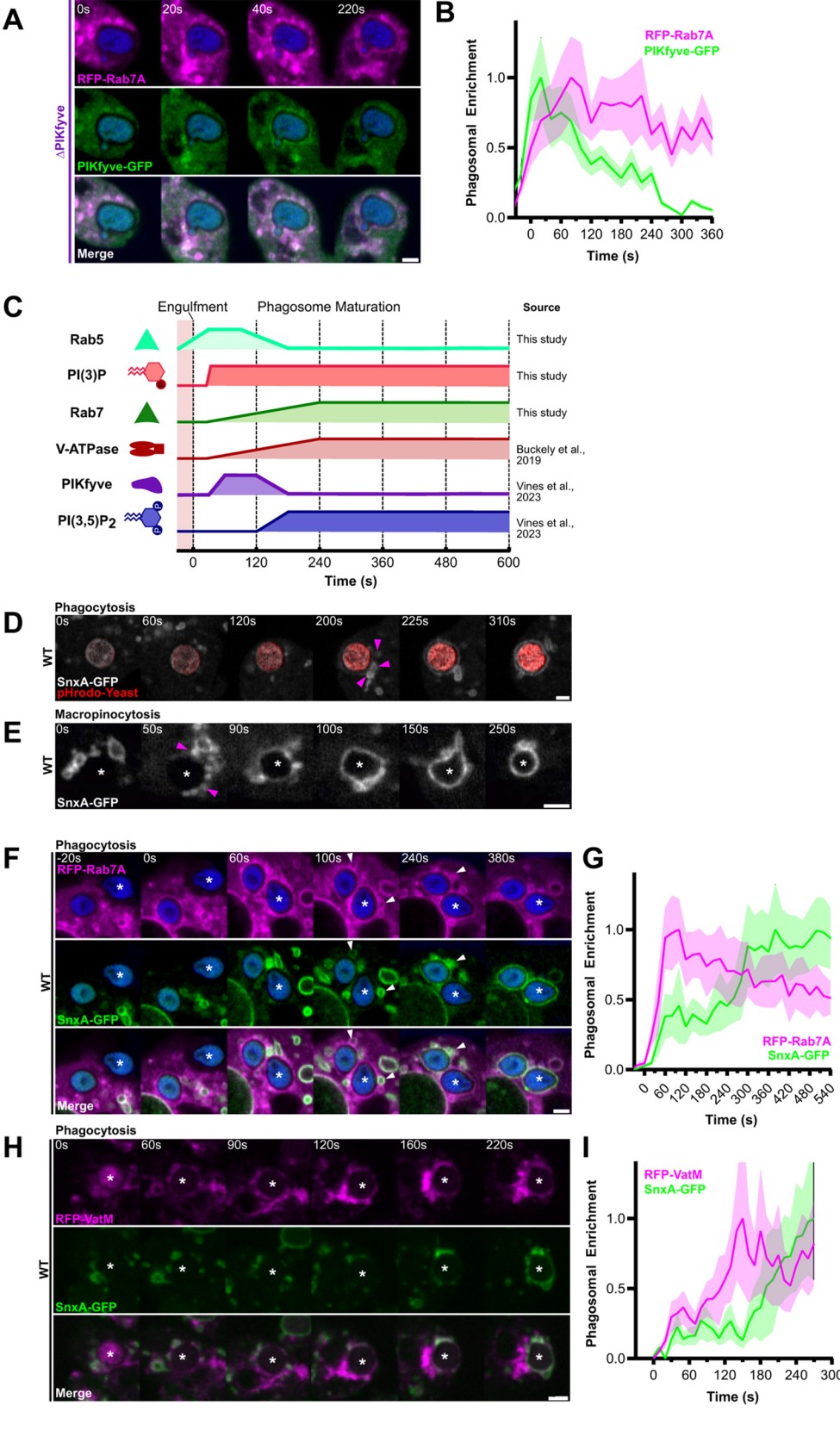

**Fig. 4. PI(3,5)P$_2$ is delivered after Rab7 to both macropinosomes and phagosomes.** (A) Timelapse of RFP–Rab7A and PIKfyve–GFP in ΔPIKfyve cells (rescued). (B) Fluorescence intensity around the phagosome in A. (C) Summary of phagosome maturation dynamics determined from this and previous studies. When the line is low, the respective component is absent from the phagosomal membrane, when the line is high the component is enriched. (D,E) Timelapses of SnxA–GFP recruitment following either phagocytosis of a pHrodo-yeast (red) or macropinosomes formation, respectively. Pink arrowheads indicate clustering SnxA vesicles. See Movie 6 for full sequence. WT, wild type. (F,G) Timelapse of RFP–Rab7A (magenta) and SnxA–GFP (green) during phagocytosis of Alexa Fluor 405-yeast (blue) in wild-type cells. Arrowheads indicate clustered vesicles marked by both SnxA and Rab7. Fluorescence intensity around this phagosome is shown in G. (H,I) Timelapses of RFP–VatM (magenta) and SnxA–GFP (green) during phagocytosis in wild-type cells. Fluorescence intensity around phagosome in H is shown in I. For all intensity graphs, data is mean±s.d. intensity of a linescan along the respective phagosomal membrane normalised between minimum and maximum intensities observed. Images representative of more than five experimental repeats. Scale bars: 2 μm.

marked by SnxA, as was evident in the quantification of intensity (Fig. 4G). Co-expression of SnxA–GFP alongside RFP–VatM yielded similar results, with SnxA vesicles representing a later-fusing subset of VatM-positive endosomes (Fig. 4H,I).

Together, our data demonstrate the presence of a Rab7/VatM/PI(3,5)P$_2$-positive compartment which is delivered to early phagosomes following PIKfyve recruitment, approximately 120 s post-engulfment. This compartment is separate from an additional

population of Rab7/V-ATPase-positive endosomes which cluster earlier and do not contain PI(3,5)P$_2$. This indicates the sequential delivery of lysosomal components and depends on PIKfyve for its regulation.

## Some of the clustering PI(3,5)P$_2$-positive vesicles are macropinosomes

We next investigated the origin of the pool of Rab7/V-ATPase/PI(3,5)P$_2$-positive vesicles that fuse with phagosomes. Our previous work shows the primary compartment marked by SnxA in *Dictyostelium* and mammalian cells are macropinosomes (Vines et al., 2023) (Fig. 4E) and this temporally overlaps with the accumulation of Rab7 we show here (Fig. 3A). Others have also shown that macropinosomes can undergo homotypic fusion (Dolat and Spiliotis, 2016; Hamasaki et al., 2004; Neuhaus et al., 2002; Tu et al., 2022); using sequential pulses of different coloured dextrans, we were also able to observe macropinosomes fusing with one another from relatively early stages of maturation (Fig. S2B).

Therefore, we hypothesised that the Rab7-, V-ATPase- and PI(3,5)P$_2$-containing macropinosomes fuse with phagosomes in a PIKfyve-dependent manner.

To test this, we labelled macropinosomes by incubating cells with a 5 min pulse of 70 kDa dextran, before washing and addition of yeast. In this way, we were able to follow interactions between a defined population of dextran-filled macropinosomes and nascent phagosomes. Using cells co-expressing RFP–2xFYVE and SnxA–GFP, we observed that many of the SnxA-positive vesicles clustering around nascent phagosomes also contained dextran and were positive for RFP–2xFYVE (white arrowheads, Fig. 5A; Movie 7). We have previously shown that both lipids are only present on macropinosomes at between 3 and 8 min of maturation (King and Kay, 2019; Vines et al., 2023). This matches the timings of our pulse–chase experiment and indicates macropinosomes cluster around phagosomes at a defined point in their maturation.

In addition to SnxA-positive macropinosomes, we also observed additional populations of SnxA–GFP-negative macropinosomes,

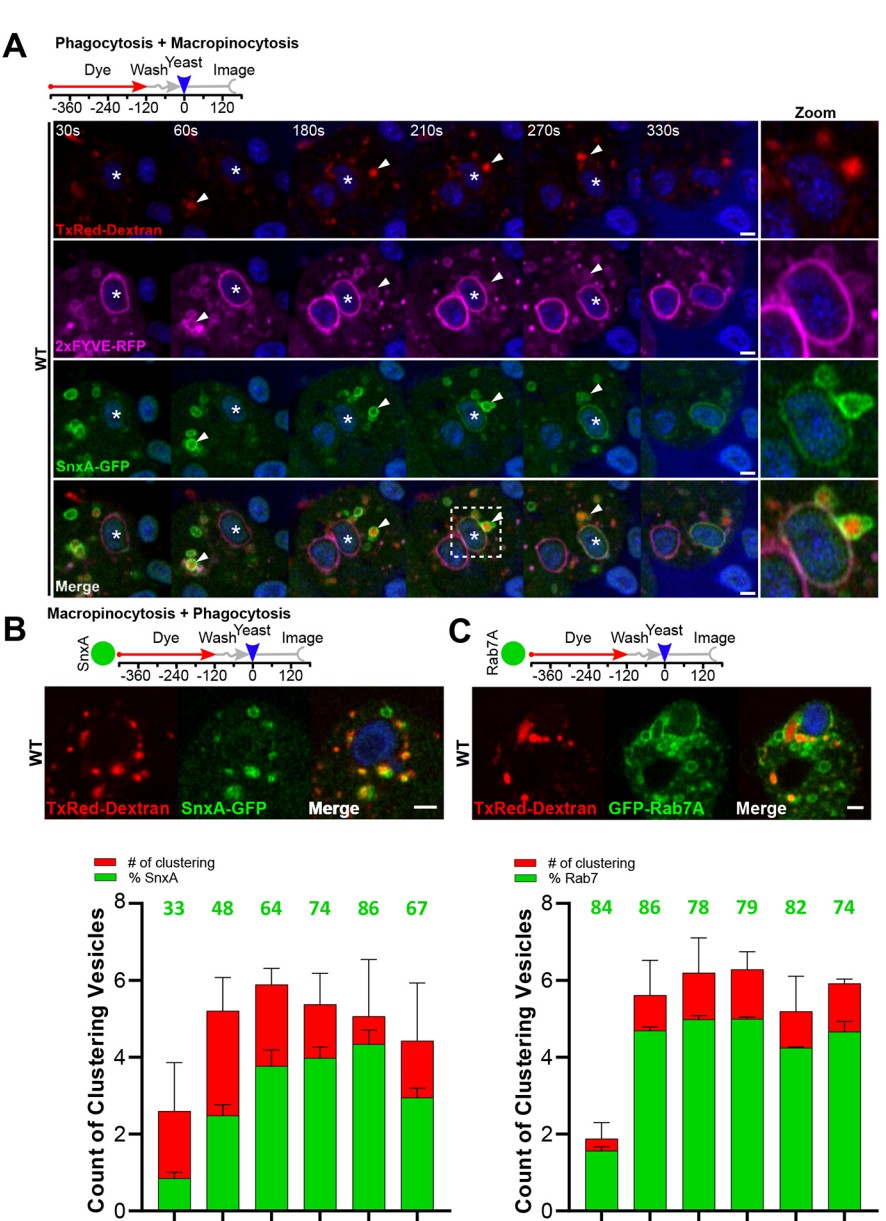

**Fig. 5. Rab7-, PI(3)P- and PI(3,5)P$_2$-positive compartments cluster around macropinosomes.** (A) Timelapse of cells co-expressing RFP–2xFYVE (magenta) and SnxA–GFP (green) during phagocytosis of Alexa Fluor 405-yeast (blue) in wild-type (WT) cells. Schematic shows experimental procedure. Cells were pre-incubated for 5 min with a 70 kDa TxRed–dextran (red) to load macropinosomes before dextran washout and addition of yeast. White arrowhead indicates a 2xFYVE- and SnxA-positive macropinosome that docks and fuses with the phagosome during acquisition of SnxA to the phagosomal membrane. See Movie 7 for full time series. Images representative of more than five experimental repeats. (B,C) Images and quantification of clustering macropinosome identity. Wild-type cells expressing GFP–Rab7A or SnxA-GFP were treated with a dextran pulse-chase before phagocytosis of yeast. Macropinosomes within 1 µm of the phagosome were identified at each timepoint and both total number (red bars) and proportion colocalizing with the respective GFP-reporter (green bars and numbers) were scored manually. Frames were binned into 2-min groups, data shown are the mean±s.e.m. of three independent experiments. Scale bars: 2 µm.

which clustered earlier. To determine the identities of these macropinosomes and how they changed over time, we imaged macropinosome clustering in cells expressing different GFP fusion reporters as above. At 2 min intervals post-phagocytosis, all dextran-labelled macropinosomes within 1 µm of the phagosomal membrane were scored for the presence of each reporter (Fig. 5B,C). In each experiment, the number of phagosome-proximal macropinosomes started low (initially about two per phagosome) but increased to about five per phagosome by 2 min, indicating temporal regulation of the interaction between the two compartments.

Consistent with our timelapse imaging of SnxA–GFP delivery, only 33% of proximal macropinosomes were positive for SnxA–GFP at initial timepoints. Based on the timelapse experiments above, this likely indicates the background false-positive rate for this method caused by macropinosomes that are near phagosomes by

chance. This increased to 80% at 6 and 8 min, consistent with the SnxA–GFP-positive macropinosome docking observed above. In contrast, at all timepoints 80% of proximal macropinosomes were positive for GFP–Rab7A. Therefore the increased population of macropinosomes that actively clusters around phagosomes at ~5 min is defined by PI(3,5)P$_2$, rather than Rab7.

## PIKfyve is required for fusion of macropinosomes with phagosomes

As the clustering of macropinosomes correlates with the presence of PI(3,5)P$_2$, we tested whether this was perturbed by PIKfyve disruption. For this, we repeated the dextran and yeast pulse-chase described above in both wild-type and ΔPIKfyve cells. As before, in wild-type cells, macropinosomes began clustering around phagosomes from within 30 s of engulfment (Fig. 6A). This was completely lost in

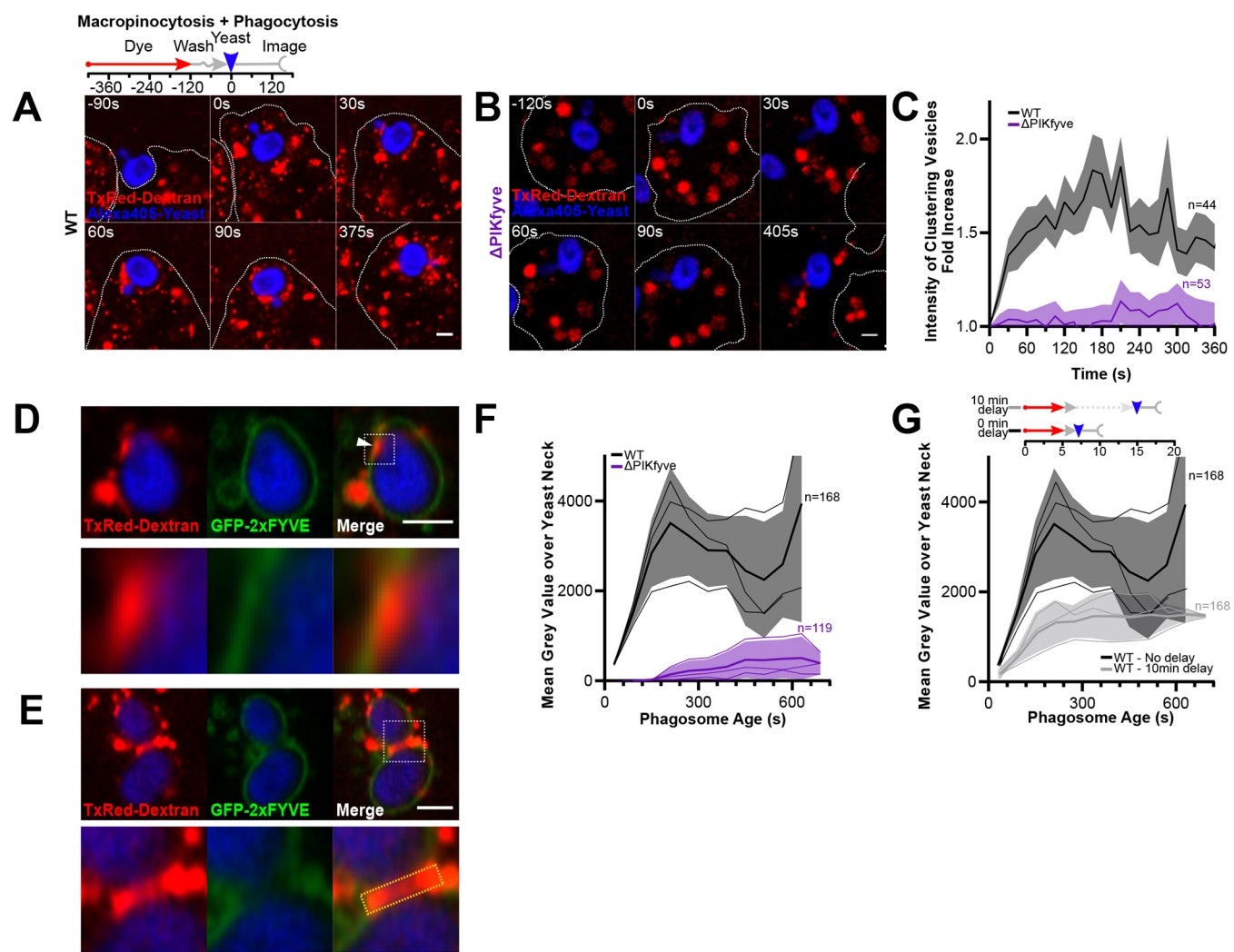

**Fig. 6. PIKfyve is required for macropinosome-phagosome fusion.** (A,B) Timelapses of interactions between early macropinosomes (red) and phagosomes (blue) in wild-type (WT) and ΔPIKfyve cells, after sequential addition of fluorescent dextran and yeast. Schematic shows experimental procedure. Macropinosomes cluster around phagosomes in wild-type but not in ΔPIKfyve cells. Whited dotted lines show cell outlines. (C) Quantification of dextran intensity within 0.5 µm of the phagosome, averaged over multiple events. Graph shows the mean±s.e.m. fold enrichment normalised to 10 s before engulfment. All values for ΔPIKfyve cells after 15 s are statistically significant from control cells (P<0.005, two-way ANOVA with multiple comparisons). (D) TxRed–dextran (red) accumulation within the phagosomal membrane (green) marked by 2xFYVE (arrowhead). (E) As D but using a budded yeast. Note strong dextran accumulation around the bud neck (dashed box). (F) Quantification of mean±s.e.m. dextran intensity in this area averaged across multiple events binned by the age of the phagosome. All values for ΔPIKfyve cells after 150 s are statistically significant from control cells (P<0.005, two-way ANOVA with multiple comparisons). (G) Quantification of dextran delivery (as in F) after a 10-min delay before addition of yeast (see schematic, same control data as in F). Older macropinosomes show less delivery to phagosomes. Scale bars: 2 µm.

ΔPIKfyve cells, where the enlarged macropinosomes were never clearly observed interacting with the phagosome (Fig. 6B). We quantified this by measuring the average dextran signal within 0.5 μm of the yeast, which indicated that this failure to cluster was persistent throughout maturation (Fig. 6C).

To determine whether macropinosomes were just clustering around phagosomes or fusing with them, we also examined delivery of dextran to the phagosomal lumen. Using GFP–2xFYVE to highlight the bounding phagosomal membrane, it was occasionally possible to observe dextran accumulation in small pockets that appeared to be within the phagosomal envelope (arrow, Fig. 6D). This was difficult to reliably quantify, but we noticed that when budding yeast were engulfed, dextran preferentially accumulated within the bud neck region of the phagosome due to its negative curvature (Fig. 6E). Using budded yeast therefore enabled us to reproducibly observe and quantify dextran delivery within the phagosome. This demonstrated fusion of early macropinosomes to yeast-containing phagosomes in wild-type cells, which was completely lost in ΔPIKfyve cells (Fig. 6F). We also tested whether macropinosomes fused with bacteria-containing phagosomes by addition of RFP-expressing *E. coli* after a 5-min dextran pulse. Consistent with our yeast data, fluorescent dextran was clearly observed in bacteria-containing phagosomes, trapped between the bacterial surface and phagosomal membrane (Fig. S4C,D). Although the small size prevented accurate quantification of this over time and multiple events, we never observed dextran within bacterial phagosomes in ΔPIKfyve cells. Therefore, PIKfyve is required for both clustering and fusion of macropinosomes to early phagosomes independent of their cargo and size.

As both PI(3)P and PI(3,5)P$_2$ are only present on macropinosomes for the first 7–8 min of maturation (Vines et al., 2023), we asked whether macropinosomes were only fusion-competent over this period. To test this, cells were again exposed to a 5 min pulse of dextran, but were then incubated for 10 min for the labelled macropinosomes to mature to a later stage before addition and phagocytosis of yeast. This delay significantly decreased accumulation of dextran at the yeast neck with only ∼20% of the dextran delivery remaining (Fig. 6G). As macropinosomes also fuse with each other (Fig. S2B), this residual macropinosome–phagosome fusion can potentially be attributed to older macropinosomes fusing to newer macropinosomes, which are then able to fuse with the nascent phagosome. Macropinosomes are therefore only competent to fuse with phagosomes at a specific early stage (2–8 min old), when they possess both PI(3)P and PI(3,5)P$_2$.

Our data demonstrate a specific role for PIKfyve in the delivery of macropinosomes to newly formed phagosomes. These macropinosomes are enriched in V-ATPase, Rab7 and lysosomal enzymes, all of which fail to accumulate on or in phagosomes in ΔPIKfyve cells (Buckley et al., 2019). Fusion with macropinosomes therefore provides a mechanism for delivery of lysosomal components to early phagosomes. However, many phagocytic cells, such as bacterially grown *Dictyostelium* do not undertake as high levels of macropinocytosis as the axenically grown cells used above. We therefore tested whether PIKfyve is important for phagosomal proteolysis in cells undertaking less macropinocytosis. As previously reported macropinocytosis was reduced by >80% in both wild-type and *PIKfyve−* cells when grown on bacteria (Fig. S4C) (Williams and Kay, 2018b). This had no effect on phagosomal proteolysis, which was still almost undetectable in the absence of PIKfyve (Fig. S4D). Therefore, although macropinosomes represent one class of vesicles that fuse in a PIKfyve-dependent manner, PIKfyve is important for phagosomal maturation even in cells with low levels of macropinocytosis, indicating additional PIKfyve-dependent fusion events. PIKfyve therefore provides a general mechanism to drive heterotypic fusion with phagosomes with physiological importance for effective bacterial killing and protection from pathogens (Buckley et al., 2016).

## DISCUSSION

In this study, we expand our understanding of the complex dynamics of Rab and phosphoinositide signalling that underpin the processing of phagosomes and macropinosomes. We describe a complex pathway in which distinct pools of endosomes are sequentially delivered to phagosomes over the first few minutes of maturation.

We identify one population of these endosomes as PI(3,5)P$_2$-positive early macropinosomes, which cluster around phagosomes before heterotopic fusion. Whereas fusion of phagosomes with lysosomes is well studied (Dayam et al., 2015, 2017; Isobe et al., 2019; Kim et al., 2014), fusion with macropinosomes is much less well understood. However, studies in both *Dictyostelium* and mammalian cells have also shown early macropinosomes fuse with one another in the first few minutes after formation, although it is unclear whether aged macropinosomes also possess this ability (Hamasaki et al., 2004; Schink et al., 2021).

Fusion between macropinosomes and phagosomes has also been observed in mammalian cells and is implicated during infection. Both *Salmonella enterica* and *Shigella flexineri* induce macropinosome formation from the ruffles formed during their entry into non-phagocytic host cells (Adam et al., 1996; Francis et al., 1993). These macropinosomes cluster around the bacteria-containing vacuole and appear to facilitate the rupture and escape of *Shigella* (Weiner et al., 2016), whereas they appear to stabilise and support the vacuolar lifestyle of *Salmonella* (Stévenin et al., 2019). Interestingly, inhibition of PIKfyve also disrupts *Salmonella* replication (Kerr et al., 2010). Although it is unclear how infection-associated macropinosomes relate to those made constitutively by professional phagocytes like macrophages or *Dictyostelium*, this highlights a common role for macropinosome delivery in the remodelling of microbe-containing compartments.

The observation that macropinosomes undergo continuous retrograde fusion challenges the view of maturation as a linear process. Our data indicate a cyclic model whereby nascent macropinosomes fuse with a slightly older population, distinguished by PIPs such as PI(3,5)P$_2$. A proportion of these will again fuse with the next round of macropinosomes, whereas the rest presumably progress further and lose fusogenicity to enter a terminally digestive phase (Fig. 7). We propose that these retrograde fusion events provide a mechanism to rapidly deliver a large volume of hydrolases, V-ATPase and other digestive machinery to newly internalised vesicles and might be particularly important for phagosomes containing large cargoes, such as the yeast, due to their large size and low surface area-to-volume ratio.

This study shows that PIKfyve is crucial for efficient Rab7 accumulation on phagosomes. In wild-type cells, Rab7A-positive vesicles continually clustered around the surface of nascent phagosomes, with Rab7A intensifying on the phagosomal membrane gradually over several minutes. This indicates that on *Dictyostelium* phagosomes at least, a significant proportion of Rab7 accumulates by fusion with Rab7-positive compartments. This is consistent with recent work (Tu et al., 2022), which also describes fusion of Rab7-positive vesicles with early-Rab5-positive macropinosomes, but contrasts with the canonical view of Rab7 recruitment from a cytosolic pool via the activities of

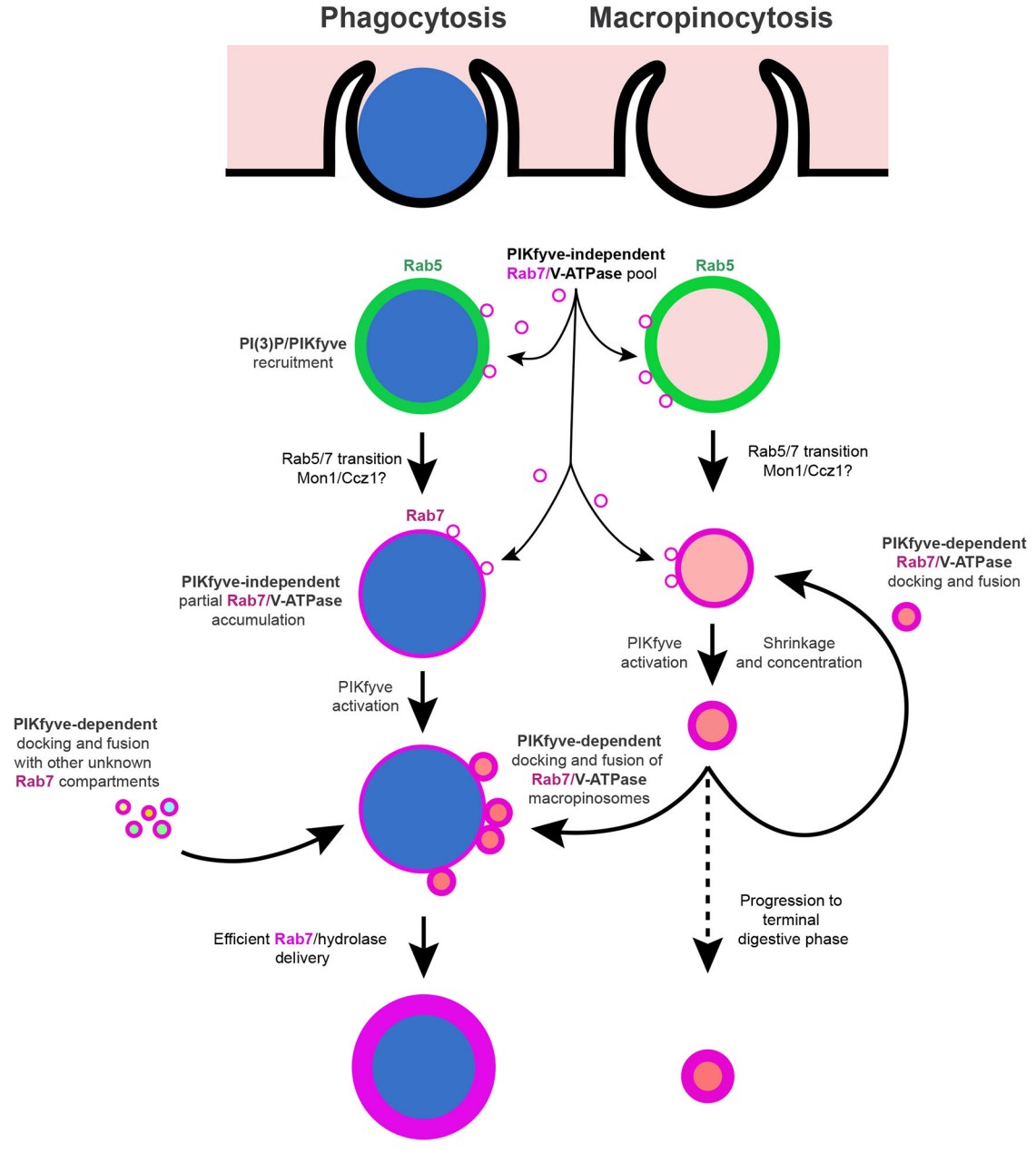

**Fig. 7. Model for the first 2 min of phagosome maturation.** The left side indicates the maturation of phagosomes in wild-type cells, with PIKfyve-deficient cells on the right. Macropinosome maturation in the centre. Wild-type phagosomes accumulate each of the proteins and lipids indicated as per Fig. 5G, with PIKfyve becoming activated ~1 min after recruitment, at the point that macropinosomes of a similar age fuse and PIKfyve itself dissociates. In the absence of PIKfyve, macropinosomes do not shrink as efficiently, and do not cluster or fuse with phagosomes.

the Mon1–Ccz complex (Langemeyer et al., 2020). Whereas we clearly identify macropinosomes as one source, it is clear that other as-yet-unidentified compartments also require PIKfyve to fuse with phagosomes and account for a significant proportion of the acidification and proteolysis observed (Buckley et al., 2019).

Our data indicate both PIKfyve-dependent and independent pools of Rab7, and although large yeast-containing phagosomes retain only ~15% of the GFP–Rab7 signal on their surface in the absence of PIKfyve, this appears to be much less reduced in phagosomes containing bacteria, which still acquire comparable GFP–Rab7 levels to wild-type cells, just more slowly. Although we cannot exclude that bacteria-containing phagosomes mature differently, this is consistent

with a limited PIKfyve-independent Rab7 pool that gets more limiting as phagosomal size increases. In the absence of obvious vesicles clustering around phagosomes in PIKfyve mutants, it is likely that this small pool of PIKfyve-independent Rab7 comes from a cytosolic pool via classical Mon1–Ccz1-mediated Rab5-to-Rab7 exchange. Although Mon1 and Ccz1 have previously been implicated in early phagosome maturation in *C. elegans* (Kinchen and Ravichandran, 2010), we were unable to visualise their *Dictyostelium* orthologues, so their role in this system remains undefined. To further complicate matters, we also observe a pool of V-ATPase and Rab7-positive vesicles that cluster around and presumably fuse with phagosomes immediately after engulfment, before PIKfyve or PI(3,5)P$_2$ can be

seen (Movie 1 and see Buckley et al., 2019). These very early fusing compartments appear to lack PI(3)P (Fig. 5A, and see Vines et al., 2023), so are unlikely to be early endosomes. Nor are these clustered vesicles consistent with the dynamics of WASH or Retromer (Buckley et al., 2016). Therefore, a further subpopulation of PIKfyve-independent Rab7/V-ATPase vesicle fusions might exist, identification of which requires further work.

How might PIKfyve regulate fusion with phagosomes? In mammalian cells, fusion of macropinosomes with lysosomes is at least partially regulated by recruitment of the septin cytoskeleton in a PIKfyve-dependent manner (Dolat and Spiliotis, 2016). However, *Dictyostelium* do not possess septins, and we find no defect in macropinosome maturation in these cells, indicating that fusion of phagosomes with lysosomes uses an alternative PIKfyve-dependent mechanism. A strong alternate candidate is the homotypic fusion and protein sorting (HOPS)-tethering complex. In yeast, the Rab7 homologue, Ypt7, directly interacts with the HOPS complex, which itself mediates endosomal fusion (Brett et al., 2008; Ostrowicz et al., 2010). A similar, less direct, interaction between Rab7 and HOPS occurs in higher eukaryotes through the Rab7 effector RILP (Lin et al., 2014; van der Kant et al., 2013). In *Dictyostelium*, disruption of Rab7A also results in severe defects in lysosomal activity and delivery of immature lysosomal enzymes (Rupper et al., 2001). It is possible that this tethering or fusion activity somehow requires PI(3,5)P$_2$, but this remains unexplored. It is also possible that PIKfyve in some way regulates Rab7 activation, rather than just recruitment, but we have been unable to assess this with the current available tools.

Another potential mechanism for the role of PIKfyve in membrane fusion could be through the activity of phagosomal ion channels. PI(3,5)P$_2$ has been shown to activate Ca$^{2+}$ ion channels on phagosomal membranes, and in mammalian cells, overexpression of the PI(3,5)P$_2$-activated Ca$^{2+}$ ion channel TRPML1 partially rescues the characteristic ΔPIKfyve swollen endosome phenotype (Dong et al., 2010). Whether a reduction in swollen endosomes would also restore heterotypic fusion with macropinosomes remains to be tested, although the *Dictyostelium* TRPML1 homolog (MLCN1), does not localize to phagosomes until after post-lysosome transition (Lima et al., 2012), meaning there must be a slightly different mechanism in this organism.

Although the mechanism(s) remains elusive, we clearly observe delivery of multiple pools of Rab7 to phagosomes, indicating a more complex picture than the canonical view of fusion with a single lysosomal population. Analyses of *Dictyostelium* phagosomes purified at different stages of maturation also show that different proteases are delivered and recovered over time (Gotthardt et al., 2006; Souza et al., 1997). The timescale in our experiments is much shorter, but both studies suggest that multiple endosomal populations are involved in phagosome maturation. The sequential fusion of different endosomal and lysosomal populations was also observed in studies of macropinosome maturation in macrophages (Racoosin and Swanson, 1993) indicating that this is likely a universal phenomenon.

A final interesting aspect of our studies is the clear differences between phagosome and macropinosome maturation in *Dictyostelium*. Although early steps such as the Rab5-to-Rab7 transition and acquisition of PI(3)P followed by PI(3,5)P$_2$ appear identical, only phagosomal degradation is strongly affected by loss of PIKfyve. In contrast, PIKfyve-deficient macropinosomes digest normally and still acquire Rab7, even though their shrinkage is reduced – similar to observations in mammalian cells (Freeman et al., 2020; Kerr et al., 2010). PIKfyve therefore appears to play independent roles in shrinkage and degradation, and in *Dictyostelium* at least, only phagosomes require PIKfyve for Rab7

and lysosomal delivery. These differences are likely driven by signalling from phagocytic receptors, but how this is mediated and how it functionally affects killing and digestion remain unclear.

Many aspects of maturation are shared between phagosomes, macropinosomes and other endocytic pathways, such as classical clathrin-mediated endosomes. These also transition from a Rab5- and PI(3)P-positive early form to later compartments demarked by Rab7, V-ATPase and lysosomal components. Degradation of classically endocytosed receptors, autophagosomes and entotic vesicles are also sensitive to PIKfyve inhibition (de Lartigue et al., 2009; Krishna et al., 2016; Qiao et al., 2021). It is therefore likely that at least some of our observations on phagosomes are applicable to other endocytic routes, although the differences with macropinosomes indicate previously unexpected complexity. Nonetheless, in this study, we further define the complex sequence of events that occur in the first minutes of a phagosome, and describe key insights into how this is regulated by PIKfyve and PI(3,5)P$_2$.

## MATERIALS AND METHODS

### *Dictyostelium* culture
All *Dictyostelium discoideum* cells were derived from the Ax2 (Kay) laboratory strain background (from the *Dictyostelium* stock centre, strain ID: DBS0235521) unless stated otherwise and grown in adherent culture in filter sterilised HL5 medium (Formedium) at 22°C. Bacterially grown cells were cultured for 48 h prior to the experiment in KK2 buffer (0.1 M potassium phosphate, pH 6.1) with *Klebsiella aerogenes* as described previously (Paschke et al., 2018), with the bacteria removed by three washes in KK2 prior to experiments. Cells expressing extrachromosomal plasmids were transformed by electroporation and grown in appropriate antibiotic selection by addition of either 20 µg/ml hygromycin (Invitrogen) or 10 µg/ml G418 (Sigma). The PIKfyve knockout strain in Ax2 background was previously described (Buckley et al., 2019). Live-cell imaging was performed in defined SIH medium (Formedium).

### Molecular biology
All gene sequences used for cloning were obtained from dictybase (http://www.dictybase.org/) (Fey et al., 2013). The generation of SnxA–GFP (pJSK619), PIKfyve–GFP (pJV0025) and integrating SnxA–GFP plasmids has been described previously (Vines et al., 2023). GFP–Rab5A (pJV0054) was generated similarly; cDNA was cloned via PCR using primeSTAR max DNA polymerase (Clontech). This was followed by subcloning into a zero blunt TOPO II vector (Life Technologies), before cloning into the appropriate BglII/SpeI sites of the pDM expression plasmid (Paschke et al., 2018; Veltman et al., 2009). The coding sequence was confirmed by restriction digest. GFP–2xFYVE (pJSK418) uses the sequence from the human *HRS* gene and was previously described (Calvo-Garrido et al., 2014). GFP–Rab7A expression plasmid (pTH70) was kindly gifted by Huaqing Cai, Institute of Biophysics, Chinese Academy of Sciences, Beijing, China (Tu et al., 2022). Co-expression plasmids were created by first cloning the RFP fusion sequence into the pDM series shuttle vectors (pDM1042 and pDM1121; gifts from Douwe Veltman, Beatson Institute for Cancer Research, Glasgow, UK), before cloning as an NgoMIV fragment into the appropriate GFP expression vector.

### Preparation of fluorescent yeast and dextran
*Saccharomyces cerevisiae* were used for phagocytosis assays. To obtain non-budded yeast, cells were grown for 3 days at 37°C in standard YPD medium (Formedium) until in stationary phase. For a budded population, growth was halted while yeast were still in log phase. Both populations were then centrifuged at 1000 *g* for 5 min and resuspended to a final concentration of 10$^9$ cells/ml in PBS pH7, then frozen until use. Yeast were fluorescently labelled using either pHrodo red succinimidyl ester (Life Technologies), or Alexa-Fluor-405 succinimidyl ester (Life Technologies) at a final concentration of 0.25 mM and 2.5 mM, respectively. 0.5×10$^9$ yeast were resuspended in 200 µl PBS at pH 8.5 and incubated with 10 µl of prepared dye for 30 min at 37°C with gentle shaking. After, yeast were pelleted (1000 *g* for 5 min) and sequentially washed in 1 ml of PBS pH 8.5,

1 ml of 25 nM Tris-HCl pH 8.5, and 1 ml of PBS pH 8.5 again. Finally, yeast were resuspended in 500 µl of KK2 pH 6.1 and kept at −20°C. Yeast were diluted in SIH to a working concentration of $10^8$ cells/ml before use.

Texas-Red–70 kDa dextran (Life Technologies) was resuspended in water and diluted to a working concentration of 2 mg/ml. Far-Red dextran was generated by labelling 70 kDa dextran (Invitrogen) with Alexa Fluor 680 NHS Ester (Invitrogen) as above, removing unbound dye by dialysis in KK2.

### Microscopy and image analysis

Approximately $10^6$ *Dictyostelium* cells were seeded into 35 mm microscopy dishes with glass bottoms (MatTek P35G-1.5-14-C) for fluorescence microscopy and left to grow overnight in SIH medium (Formedium). Imaging was conducted using a Zeiss LSM880 AiryScan Confocal microscope equipped with a 63× Plan Apochromat oil objective.

To perform dual-colour timelapse phagocytosis assays, most of the medium was removed just prior to imaging, and 20 µl of $10^8$ dyed yeast added. After 60 s, a thin layer of 1% agarose in SIH was overlaid on the cells, and excess medium was removed. For normal timelapses, images were captured every 10 s for up to 10 min. For extended timelapses, images were captured every 60 s. Fluorescence intensity analysis around each phagosomal membrane was carried out automatically using the Python plugin pyimagej, which has been described previously (Vines et al., 2023). In brief, phagosomal contents were identified and segmented by using the fluorescent yeast, which are large and easy to follow. The fluorescence channel containing the yeast was thresholded, and large yeast-sized particles identified to be followed over time by examining similar particles on adjacent frames. This allowed individual phagosomes to be followed over the course of a few minutes. Increases in membrane fluorescence were then calculated by expanding the particle perimeter and taking average fluorescence readings in a banded area.

To perform dextran pulse-chases, the majority of the medium was removed and replaced with 50 µl of 2 mg/ml 70 kDa Texas Red dextran for the specified duration. Dextran was then removed by washing three times in SIH medium, leaving ∼1 ml of SIH in the dish. Cells were immediately imaged, with multiple fields of view captured every 2 min for 10 min. Quantification of macropinosome number and size was quantified using an automated ImageJ script. GFP colocalization at each time point was scored manually.

For macropinosomes–phagosome fusion assays, 50 µl of 2 mg/ml Texas Red dextran was added to cells for the time indicated for macropinosomes to form. The dish was then washed as above, before addition of 20 µl of $1×10^8$ dyed yeast and then agarose overlay after a further 60 s. Clustering analysis was performed using a modified version of the automated analysis pipeline described above. For macropinosome–macropinosome mixing a second dye, 50 µl of Alexa-Fluor-680 70 kDa Dextran, was added instead of yeast. For macropinosome–phagosome fusion assays, budded yeast were used instead and analysis of fusion performed manually in ImageJ by measuring the mean intensity across the yeast neck over time. For analysis of clustering macropinosomes, macropinosomes within 1 µm of the phagosome were manually scored by a researcher (owing to the obvious endosomal morphology of the mutant cells, experimental conditions would have been apparent) as GFP-positive or -negative, and binned into 2-min intervals.

### Proteolysis measurements

Phagosomal proteolysis was measured by the incubating the cells with DQ-BSA and Alexa Fluor 594 conjugated to 3 µm silica beads as described previously (Sattler et al., 2013). Briefly, $3×10^5$ cells in LoFlo medium (Formedium) were seeded in triplicate wells of a 96-well plate to form a monolayer and left for 2 h. $1.5×10^5$ beads were then added to each well and briefly centrifuged onto the cells, and fluorescence measurements made over time using a plate reader. The DQ-BSA signal was normalised to that of Alexa Fluor 594 to account for phagocytosis rates.

Proteolysis within macropinosomes was performed by measuring the increase in fluorescence upon digestion of uncoupled DQ-BSA. Cells were seeded as above and exposed to a 2-min pulse of 0.2 mg/ml DQ-BSA before washing thrice in LoFlo. Fluorescence was then measured every minute at 495 nm excitation and 520 nm emission using a plate reader.

### Macropinocytosis assays

Macropinocytosis was measured by the uptake of 70 kDa TRITC–dextran (Life Technologies) using flow cytometry as previously described (Williams and Kay, 2018a). Briefly, $5×10^4$ cells were seeded in 50 µl HL5 in duplicate wells of a 96-well plate for each time point and left 2 h to recover, before addition of 50 µl of HL5 containing 1 mg/ml TRITC–dextran, staggered for each time point. After 90 min, the media was tipped off, the plate was washed once in ice-cold KK2, then ice cold KK2 with 5 mM sodium azide added to each well to detach. The TRITC signal in each sample was then measured using an Attune NxT flow cytometer using an autosampler. The signal at 0 min was then subtracted from each sample and normalised to the maximum signal in wild-type cells grown in HL5.

### Western blotting

Wild-type and ΔPIKfyve cells expressing GFP–Rab5A, GFP–2xFYVE or GFP–Rab7A were analysed by SDS-PAGE and western blotting using a rabbit anti-GFP primary antibody (a gift from Andrew Peden; School of Biosciences, University of Sheffield, UK; 1:1000) and a fluorescently conjugated anti-rabbit-IgG 800 (Life Technologies, cat. no. A32735; 1:10,000), using standard techniques. The endogenous biotinylated mitochondrial protein methylcrotonoyl-CoA Carboxylase 1 was used as a loading control using Alexa Fluor 680-conjugated streptavadin (Davidson et al., 2013).

### Statistics

Graphpad Prism 9 was used for statistical analysis. The figure legends provide details on the number of biological replicates and the statistical tests used for each experiment. For experiments where individual phagocytic events are followed over time over multiple movies across at least 3 days, individual cells are classed as biological units and error bars indicate standard deviation. In experiments where multiple cells are measured in parallel (e.g. pulse–chase) biological repeats were performed on separate days, and error bars indicate s.e.m. of each independent experiment. The statistical tests used are described in each figure legend. $P<0.05$ was considered significant, and is indicated as *$P>0.05$, **$P>0.01$ and ***$P>0.005$ throughout.

### Acknowledgements

The authors thank Huaqing Cai for Rab expression plasmids, Douwe Veltman and Peggy Paschke for their continued development of his expression system and Robert Insall for his support in the early days of this project. Imaging work was performed in the Wolfson Light Microscopy Facility at Sheffield.

### Competing interests

The authors declare no competing or financial interests.

### Author contributions

Conceptualization: J.H.V., C.B., J.S.K.; Data curation: J.H.V., J.S.K.; Formal analysis: J.H.V., I.W., J.S.K.; Funding acquisition: J.S.K.; Investigation: J.H.V., C.B., I.W., D.S., J.S.K.; Methodology: J.H.V., C.B., D.S., J.S.K.; Project administration: J.S.K.; Resources: J.S.K.; Supervision: J.S.K.; Visualization: J.S.K.; Writing – original draft: J.H.V., J.S.K.; Writing – review & editing: J.H.V., C.B., I.W., D.S., J.S.K.

### Funding

J.H.V. was funded by Royal Society grant RGF\EA\180126 and Wellcome Trust Discovery Award 311591/Z/24/Z, J.S.K. is funded by a Royal Society University Research Fellowship URF\R\201036. Open Access funding provided by University of Sheffield. Deposited in PMC for immediate release.

### Data and resource availability

All relevant data and details of resources can be found within the article and its supplementary information.

### Peer review history

The peer review history is available online at https://journals.biologists.com/jcs/lookup/doi/10.1242/jcs.264814.reviewer-comments.pdf

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
