## [Peer Review File · Journal of Cell Science]

PIKfyve is required for efficient phagosomal Rab7 acquisition and the delivery and fusion of early macropinosomes to phagosomes

James H. Vines, Catherine M. Buckley, Ilona Willson, Daniel S. Stark and Jason S. King
DOI: 10.1242/jcs.264814

Editor: Robert Parton

Review timeline

Submission to Review Commons:	25 July 2023
Submission to Journal of Cell Science:	22 November 2023
Editorial decision:	24 November 2023
First revision received:	24 February 2026
Accepted:	10 March 2026

Reviewer 1

Evidence, reproducibility and clarity

PIKfyve/Fab1, a kinase responsible for phosphorylating PI3P to produce PI(3,5)P₂, regulates phagosome maturation across various organisms. A previous work from the authors' group demonstrated that disrupting PIKfyve in *Dictyostelium* inhibits the delivery of V-ATPase and hydrolase, thus dramatically reducing the ability of cells to acidify newly formed phagosomes and digest their contents. The current manuscript further dissects the function of PIKfyve and PI(3,5)P₂. Using live cell imaging, the authors show that nascent phagosomes acquire Rab7 by fusion with multiple populations of Rab7-positive vesicles, and the loss of PIKfyve abolishes this event. One of these fusogenic vesicle populations was identified as PI(3,5)P₂-positive macropinosomes, which dock and fuse with phagosomes in a PIKfyve-dependent manner. Based on these findings, the authors propose that PIKfyve contributes to phagosome maturation by promoting heterotypic fusion between phagosomes and macropinosomes, which help deliver regulatory components necessary for phagosome acidification and digestion. This study provides fresh insights into the process of phagosome maturation. The work was well designed, performed and presented, and the manuscript is clearly written. However, there are several questions that should be addressed to strengthen the conclusions of the manuscript.

Major points:

1. The observation that macropinosomes undergo retrograde fusion with newly formed phagosomes to facilitate phagosome maturation is an interesting notion that challenges the traditional model. However, not all phagocytes exhibit a high level of macropinocytosis, and axenic *Dictyostelium* cells used in the study may be an exception. Thus, it remains unclear whether fusion with macropinosomes is universally required for phagosome maturation. WT *Dictyostelium* cells or axenic cells cultured under SorMC/Ka condition (Paschke et al., PLoS One, 2018) exhibit significantly reduced macropinocytosis. The authors could examine whether the accumulation of Rab7 and V-ATPase on large-sized phagosomes is delayed in these cells. These experiments may help broaden the applicability of the authors' finding.
2. PIKfyve seems to play a specific role in the maturation of phagosomes but not macropinosomes. The differences may be driven by signaling from phagocytic receptors, as the author suggested. Alternatively, the large size of the yeast-containing phagosomes may require additional steps for efficient lysosomal delivery. The authors should consider examining whether PIKfyve is needed for the delivery of Rab7 and V-ATPase to phagosomes of comparable size to

regular macropinosomes, such as those containing *K. aerogenes* or small beads. In addition, whether the process also involves fusion between phagosomes and macropinosomes should be verified.

3. In the previous study from the authors' group (Buckley et al., PLoS Pathog, 2019), it was shown that the accumulation of V-ATPase on phagosomes begins immediately after internalization in both PIKfyve mutant and WT, although V-ATPase accumulation reaches only half of the levels seen in WT. This partial accumulation of V-ATPase differs from the almost complete absence of Rab7 recruitment found in this study, which raises the question of whether there exists yet another population of fusogenic vesicles that are positive for V-ATPase but negative for Rab7. This could be checked by simultaneously examining the dynamics of V-ATPase and Rab7 during yeast phagocytosis in the PIKfyve mutant.

****Minor points:****

1. It is unclear how the experiment in Figure 3G was conducted. If microscopic analysis was involved, the corresponding images should be included.
2. Page 11-Line 2, the sentence "there was no obvious clustering around the nascent phagosome (Figure 2D)." It is Figure 2E, not Figure 2D.
3. There is an inconsistency regarding the description of fluorescent fusion proteins. For example, both GFP (RFP)-2xFyve and 2xFyve-GFP (RFP), as well as GFP-Rab5 and Rab5-GFP, were used. Typically, placing GFP (or RFP) before a gene suggests N-terminal tagging, while placing it after the gene implies C-terminal tagging. The authors should clarify the position of the fluorescent tag and ensure consistency in their descriptions.
4. One of the videos was not referred in the manuscript or described in the Video legends. This video seems to correspond to Figure 5A, albeit with a different pseudo-color scheme.

Significance

Disruption of PIKfyve results in severe defects in phagosomal maturation across different organisms, but the underlying mechanism remains unclear. This study demonstrates that PIKfyve plays a specific role in phagosome maturation by promoting heterotypic fusion between macropinosomes and newly formed phagosomes. These fusion events provide a means for the rapid delivery of lysosomal components to early phagosomes. The study challenges the conventional model of phagosome maturation and provides novel insights into the complex dynamics involved. Nonetheless, further investigations are needed to elucidate the exact role of PIKfyve/PI(3,5)P₂ in regulating vesicle fusion and to explore whether the proposed model can be applied to other endocytic pathways or cell types.

Reviewer 2

Evidence, reproducibility and clarity

****Summary****

Vines et al. investigate in their manuscript the contribution of the lipid kinase PIKfyve on the maturation of phagosome. They follow the fate of the two main classes of identity markers of endocytic organelles: PIPs (PI(3)P and PI(3,5)P₂) and Rab-GTPases (Rab5 and Rab7) in wildtype and Δ PIKfyve *Dictyostelium discoideum* cells. To follow the two species of PIPs they utilize the established reporter-proteins GFP-2xFYVE (PI(3)P-binder) and GFP-SnxA (PI(3,5)P₂-binder) and correlate them with the appearance of fluorescently tagged Rab-GTPases on membranes of phagosomes in live-cell imaging.

They find that the deletion of PIKfyve does not alter the recruitment and behaviour of Rab5. Therefore a lack of PI(3,5)P₂ does not affect the early stages of phagosome formation. However, later stages of phagosome maturation are apparently affected by the lack of PI(3,5)P₂: the Rab-GTPase Rab7 is not localizing to the membrane of the phagosome. Closer inspection of their live cell imaging data led the authors to the conclusion that Rab7 is delivered by the fusion of Rab7-positive structures with the phagosomes in wildtype cells. This fusion seems to be dependent on PIKfyve and PI(3,5)P₂. However, surprisingly, the recruitment of Rab7 to macropinosomes or

endosomal structures is independent of PIKfyve. The authors conclude (i) that lysosomal components are delivered to phagosomes by fusion of PI(3,5)P₂-positive macropinosomes and (ii) a non-canonical delivery from Rab7 to phagosomes by fusion instead of GEF-dependent recruitment from the cytosol.

****Major comments****

Overall, the submitted manuscript of Vines et al. is of very good quality. The presented data supports mainly the conclusions that the authors draw. Methods and statistical analysis are sound and well-described. Their rationale, the description of results, and the presentation of data are easy to follow and understand. However, there are two major points that I would like to address here:

1. The authors show that deletion of PIKfyve results "in an almost complete block in Rab7 delivery to phagosomes" (page 17) indicating that the delivery of Rab7 depends on fusion with Rab7-positive structures. This would suggest that the Rab7-GEF Mon1-Ccz1 is not localized to the membrane of the phagosomes. Could the authors test for the presence of Mon1-Ccz1 in either fluorescence microscopy experiments or on purified phagosomes to exclude the possibility of a "canonical" Rab7 recruitment by its GEF? If the GEF is found on phagosomal membranes it would indicate that a Rab-transition from Rab5 to Rab7 occurs on the phagosome during maturation, but on a low level. The later fusion event might be a homotypic fusion of two Rab7-positive compartments. The observed fusion events could still deliver the bulk of Rab7 and other endolysosomal proteins to the phagosome. If the Rab7-GEF is not found on phagosomes how do the authors envision that the organelle keeps its identity? Is it solely dependent on PI(3,5)P₂? What is the fate of the Rab7-negative phagosome in Δ PIKfyve cells if Rab7 is not delivered to the membrane, is there degradation happening over longer periods of time?
2. In their abstract, the authors state that they "...delineate multiple subpopulations of Rab7-positive endosomes that fuse sequentially with phagosomes" (page 2, line 14,15). However, the data provides only evidence for V-ATPase or PI(3,5)P₂-containing structures and the authors conclude to my understanding that macropinosomes are the main source for vesicular structures fusing with phagosomes. I would ask the authors to please be clear on the identity of the "Rab7-donor"-structures throughout the manuscript. Saying that they delineate multiple subpopulations of endosomes seems to be overstated.

("OPTIONAL") Optionally, the authors could also try to clarify these structures' identity by including further colocalization studies with additional early and late endosomal marker proteins. Are they for example positive for early or late endosomal markers like EEA1, ESCRT or Retromer? How about organelle-specific SNAREs? This would give further insights into the character of the "Rab7-donor" structures and would allow to clarify if multiple subpopulations are contributing to phagosome maturation in a sequential order as stated in the abstract. As I am not an expert on Dictyostellium I can't estimate the effort that would go into such an experimental setup. However, since the time scale of the events in the cell is nicely worked out in this study, these colocalization studies would not need to be conducted as live-cell microscopy experiments.

****Minor comments****

Minor points:

1. The sentence "...which both deactivates and dissociates Rab5, and recruits and activates Rab7 on endosomes" is at least problematic as it suggests that Mon1-Ccz1 directly drives GTP-hydrolysis of Rab5 and dissociates it from the membrane. Indeed, Mon1-Ccz1 is shown to interfere with the positive feedback loop of the Rab5-GEF by interacting with Rabex (Poteryaev et al., 2010), so a rather indirect effect of Mon1-Ccz1. A GAP and the GDI are needed for Rab5 deactivation and dissociation from the membrane. How both are involved in the endosomal Rab-conversion is not clarified.
2. Signals of RFP-labeled proteins are difficult to interpret throughout the experiments. What are the structures that show a strong accumulation of red signal in Fig. 1A,B, Fig 2G and Fig4A (20sec.) If these are fluorescently labeled proteins it would suggest that most of the proteins cluster/accumulate in the cell. Can the authors provide better images?

3. On page 11 the authors state "...macropinosomes in Δ PIKfyve cells still appeared much larger. Quantification of their size and fluorescence intensity demonstrated that although macropinosomes started off the same size,...". This statement is not reflected in the data depicted in Fig. 3A,B. The size of the single labeled macropinosome appears to be larger in wildtype than in Δ PIKfyve cells from the beginning on. However, the quantification in Fig 3F is clear. So, are these bad examples in 3A,B, are they swapped or is this due to the additional expression of GFP-Rab7A? Could you please comment on the effect that the (over-)expression of GFP-tagged Rab-GTPases might have on the observations described in this paper in the discussion part?
4. In Fig. 6E it is hard to distinguish if the dextran is accumulating inside the phagosome. I would suggest conducting a 3D reconstruction of these images to allow judging if macropinosomes fused with the phagosomes or if they cluster around the neck of the phagosome.
5. In the discussion, the authors state that the small pool of "PIKfyve-independent Rab7" is "insufficient to for subsequent fusion with other Rab7A-positive compartments, further Rab7 enrichment, and lysosomal fusion." What is the rationale for this conclusion? Is it shown how many Rabs are necessary to induce a tethering and fusion event? It would be good to revise this part of the discussion also in respect of the first major point of my comments above.
6. The intention of the paragraph about phagosomal ion channels is for this reviewer somehow out of context. It is not clear to me how the authors relate this to their findings. It would be good to bring this into a broader context.

****Referees cross-commenting****

Reviewer #1 provides valid questions. Addressing them would improve the manuscript by allowing consideration if the findings only apply to Dictyostellium or is of broader interest.

I completely agree with the concern of Reviewer #3 that the data provided so far would also allow for alternative models. The authors need to include further controls to exclude Rab7 recruitment or activation by any other means than fusion.

Significance

The manuscript by Vines et al. describes a very interesting novel observation on how the organelle identity marker Rab7 is delivered to phagosomes. They propose a mechanism, the delivery of Rab7 by PIKfyve-dependent fusion events with Rab7-positive macropinosomes, which is in contrast to the canonical model that endosomal organelles gain their Rab7- identity by maturation from a Rab5-positive compartment with the help of the Rab7-GEF Mon1-Ccz1. In the proposed mechanism the lipid-kinase PIKfyve, which is also involved in cellular signaling processes, plays the key role. In this study the authors present profound live cell imaging experiments combined with pulse-chase uptake of phagosomal cargoes. The obtained data is giving surprising new insights on the order of events in the maturation of phagosomes and suggests an unprecedentedly important role for PIKfyve in the maturation process. These new insights are of broad interest to a readership interested in transport, maturation and signaling processes along the endolysosomal system as well as of interest in the perspective of pathogen invasion to host cells.

Reviewer 3

Evidence, reproducibility and clarity

In Vines et al., the authors used time-lapse imaging of Dictyostelium to investigate the spatial-temporal maturation of macropinosomes labelled with a short pulse of dextran and phagosomes using yeast particles. The phagocytes expressed fluorescent Rab5 and/or Rab7 and/or biosensors for PI(3)P using 2FYVE-GFP and PI(3,5)P2 using the authors recently disclosed SnxA. They quantified the dynamics of these probes in wild-type and PIKfyve-deleted cells. The authors provide evidence for their main observations, which are that: i) Rab5 and PI(3)P are acquired early and independently of PIKfyve on phagosomes and macropinosomes, ii) but phagosomes require PIKfyve to acquire Rab7, iii) that phagosomes acquire Rab7 by fusing with Rab7-containing vesicles that cluster around the phagosome, iv) that macropinosomes do not require

PIKfyve for Rab7 acquisition, and v) that PI(3,5)P₂ on phagosomes follows Rab7. While the imaging data is high quality and supports several of the claims, the major discovery as proposed here is not fully supported by the data provided. I think the authors must address the following to strengthen their otherwise beautiful work.

****Major Comments:****

1. Based on the current data, I am not entirely convinced that Rab7 is delivered mostly by fusion with other compartments. At least the data as provided cannot exclude other models. For example, Rab7-containing organelles that cluster with phagosomes may form contact sites that provide a local environment to load cytosolic Rab7. There's also a possibility that some of their Rab7 clusters are membrane sub-domains and not vesicles. Or perhaps, there is a first wave of cytosolic Rab7 recruitment, which then initiates fusion with Rab7 compartments, i.e., there is a two-phase Rab7 recruitment. While this last possibility is consistent with recruitment of Rab7 by fusion (the second phase), the authors present a model that is too simplistic and conclusive based on the data. The authors may be right, but they need to strengthen their evidence towards their claim. Maybe EM could help determine some of these issues. Perhaps better would be the use of FRAP, photo-activation, or optogenetics of Rab7. For example, if Rab7 is acquired on phagosomes after photobleaching clusters of Rab7, this would suggest a cytosolic Rab7 contribution, and if not, this would suggest their model. I recognize that these experiments are not necessarily trivial, but either the authors augment their data (as suggested or with other approaches) or significantly pare down their conclusions.
2. The focus in their manuscript is loading of Rab7 on phagosomes, but there's no indication about Rab7 activation (GTP-loading). Would the RILP-C33 probe work in Dictyostelium? If not possible, the activation state of Rab7 should still be discussed. Despite Rab7 on other organelles in PIKfyve-inhibited cells, is this active or not?
3. The authors need to better address the confusing kinetics of early Rab7 recruitment, followed by SnxA (Fig. 4G, same for VatM - Fig. 4I) - which is counterintuitive if PIKfyve activity is required to recruit Rab7. How do the authors explain this? Are phagosomes prevented from acquiring Rab7 in PIKfyve deficient cells because of a defect on phagosomes or the endo-lysosomes loaded with Rab7 (but not active).
4. Their disclosure and use of statistics is incomplete and/or inconsistent, and potentially wrong in some cases. For example, the authors disclose the number biological repeats in a few experiments (Fig. 3C, F) but not in the majority. Instead, they state the number of phagosomes without indicating biological repeats (eg. Fig. 2 and others). So, it is not possible to know if their data are reproducible. Despite not indicating independent experiments in some cases, they speak of SEM, which applies to mean of means from biological repeats. In other cases, none of this is disclosed (eg Fig. 3G). Often there is no indication of what statistical test was done OR if a statistical test was done (eg. Fig. 3G, Fig. 4, etc). I would recommend the authors review the excellent resource paper published in JCB on SuperPlots to better follow statistical expectations. This is essential to improve reproducibility and confidence in their observations.
5. Early macropinosomes fuse with early phagosomes more readily than 10-min old macropinosomes. Do 10-min old macropinosomes not fuse with older phagosomes? Is this not an issue of mismatched age?

****Minor Comments****

6. It is interesting that 2FYVE-GFP stays on phagosomes for 50 min or more - this is distinct from macrophages. Please comment. Have the authors tried other PI(3)P probes to see if the same (PX-GFP).
7. Fig. 7 model: the macropinosome in the diagram seems like a dead end as depicted - is there any arrow or change that could be added to show that it doesn't just sit there in the middle? Also, the light green on yellow hurts the eyes!
8. Fig. 3F, could be converted to volume assuming macropinosomes are spheres.
9. Pg. 10, line 10 - Vps34 is Class III PI3K, not Class II.

Significance

Overall, the potential novelty of this work is the authors' proposal that phagosomes acquire Rab7

mostly by fusion with Rab7-labelled organelles rather than a cytosolic tool. This is distinct from existing models that assume phagosomes acquire Rab7 from a cytosolic pool that is loaded onto the membrane. They also suggest that PIKfyve plays a role in this process. However, as noted above, this claim needs to stronger data as the current data allows for other possible models, in my opinion.

This work is of relevance to cell biologists interested in membrane trafficking, phagocytosis, model organisms, and microscopy.

Manuscript number: RC- 2023-02097

Corresponding author(s): Jason King

1. General Statements

We thank the reviewers for their time and both thoughtful and constructive comments. Their specific points are addressed below but a general point that we would like to comment on is that in the original version it appears we did not make our model clear enough. The dogma in the field is that Rab7 is recruited to endosomes from a cytosolic pool via exchange with Rab5 (mediated by Mon1/Ccz1). Our work instead indicates that the majority of Rab7 is delivered to *Dictyostelium* phagosomes by fusion with other endocytic compartments. It was not our intention to imply there was no canonical recruitment of Rab7 from a cytosolic pool, and indeed we provide data to show this happens at a low level and discuss this in the manuscript. Nonetheless, we clearly over-stated the exclusivity of Rab7 recruitment to phagosomes via fusion at several points and our original model cartoon, and have tried to better explain or more nuanced model with multiple routes for Rab7 acquisition in this revision, including a completely redrawn model figure (Fig. 7).

2. Description of the planned revisions

Reviewer 1:

1. The observation that macropinosomes undergo retrograde fusion with newly formed phagosomes to facilitate phagosome maturation is an interesting notion that challenges the traditional model. However, not all phagocytes exhibit a high level of macropinocytosis, and axenic *Dictyostelium* cells used in the study may be an exception. Thus, it remains unclear whether fusion with macropinosomes is universally required for phagosome maturation. WT *Dictyostelium* cells or axenic cells cultured under SorMC/Ka condition (Paschke et al., PLoS One, 2018) exhibit significantly reduced macropinocytosis. The authors could examine whether the accumulation of Rab7 and V-ATPase on large-sized phagosomes is delayed in these cells. These experiments may help broaden the applicability of the authors' finding.

As our previous work (Buckley et al. PLoS pathogens 2019) demonstrated that bacterially-grown PIKfyve mutants are also defective in bacterial killing and growth it is highly likely that cells also are defective in V-ATPase and Rab7 acquisition. However, we agree that formally testing this will further support our conclusions and improve the paper and should be quite straightforward.

We will therefore co-express GFP-V-ATPase and RFP-Rab7 in both Ax2 and non-axenic cells grown on bacteria and repeat our analysis of recruitment to phagosomes - with the caveat that non-axenic cells do not phagocytose large particles such as yeast (Bloomfield et al. eLife 2015), so the imaging and quantification will be more challenging in this case.

2. PIKfyve seems to play a specific role in the maturation of phagosomes but not macropinosomes. The differences may be driven by signaling from phagocytic receptors, as the author suggested. Alternatively, the large size of the yeast-containing phagosomes may require additional steps for efficient lysosomal delivery. The authors should consider examining whether PIKfyve is needed for the delivery of Rab7 and V-ATPase to phagosomes of comparable size to

regular macropinosomes, such as those containing *K. aerogenes* or small beads. In addition, whether the process also involves fusion between phagosomes and macropinosomes should be verified.

Whilst it is possible that large size of yeast-containing phagosomes requires additional mechanisms to process them, our previous data demonstrate that PIKfyve is also required to kill much smaller bacteria such as *Klebsiella* and *Legionella* (Buckley et al. PLoS pathogens 2019). Furthermore, in this paper we also showed that loss of PIKfyve disrupts phagosomal proteolysis using 3µm beads, and showed that V-ATPase recruitment was reduced on purified phagosomes containing 1µm beads. We therefore find consistent defects on phagosomes of different size, with different cargos. Nonetheless, the experiments above, observing V-ATPase and Rab7 in cells grown on bacteria should directly address this point.

As suggested, we will also perform a dextran pulse-chase prior to addition of bacteria to test if we can observe macropinocytic delivery to bacteria-containing phagosomes - perhaps using *E. coli* as their elongated shape may help phagosome visualisation.

3. In the previous study from the authors' group (Buckley et al., PLoS Pathog, 2019), it was shown that the accumulation of V-ATPase on phagosomes begins immediately after internalization in both PIKfyve mutant and WT, although V-ATPase accumulation reaches only half of the levels seen in WT. This partial accumulation of V-ATPase differs from the almost complete absence of Rab7 recruitment found in this study, which raises the question of whether there exists yet another population of fusogenic vesicles that are positive for V-ATPase but negative for Rab7. This could be checked by simultaneously examining the dynamics of V-ATPase and Rab7 during yeast phagocytosis in the PIKfyve mutant.

We agree with the referee that there are multiple pools of V-ATPase, and we show that there is both a very early PIKfyve-independent recruitment of both V-ATPase and Rab7 as well as a later and more substantial pool delivered in a PIKfyve-dependent manner. It is clear that V-ATPase and Rab7 do not always co-localise however - the clearest example being on the contractile vacuole, which has lots of V-ATPase but no Rab7 (the large bright magenta structure in Fig 2G.).

We suspect that the dramatically reduced, but not completely absent levels of both V-ATPase and Rab7 recruitment in the absence of PIKfyve are similar, but the challenges with imaging these very small low levels means we cannot formally exclude that there is a pool of V-ATPase vesicles that lack Rab7 which fuse to very early phagosomes. Nonetheless, as we will already be looking at V-ATPase and Rab7 in PIKfyve KO's in the experiments above will also attempt to unequivocally differentiate a pool of V-ATPase positive/Rab7 negative vesicles fusing with phagosomes.

Reviewer 2:

(1) The authors show that deletion of PIKfyve results "in an almost complete block in Rab7 delivery to phagosomes" (page 17) indicating that the delivery of Rab7 depends on fusion with Rab7-positive structures. This would suggest that the Rab7-GEF Mon1-Ccz1 is not localized to the membrane of the phagosomes. Could the authors test for the presence of Mon1-Ccz1 in either fluorescence microscopy experiments or on purified phagosomes to exclude the possibility of a "canonical" Rab7 recruitment by its GEF? If the GEF is found on phagosomal membranes it would indicate that a Rab-transition from Rab5 to Rab7 occurs on the phagosome during maturation, but on a low level. The later fusion event might be a homotypic fusion of two Rab7-positive compartments. The observed fusion events could still deliver the bulk of Rab7 and other endolysosomal proteins to the phagosome. If the Rab7-GEF is not found on phagosomes how do the authors envision that the organelle keeps its identity? Is it solely dependent on PI(3,5)P2? What is the fate of the Rab7-negative phagosome in Δ PIKfyve cells if Rab7 is not delivered to the membrane, is there degradation happening over longer periods of time?

This is an excellent suggestion, for which we thank the reviewer. Mon1 and Ccz1 are highly conserved, with clear *Dictyostelium* orthologues that have never been studied. Our model is that there is a small proportion of Rab7 driven by this canonical pathway so would expect Ccz1/Mon1 to coincide with loss of Rab5 and be unaffected by loss of PIKfyve - although subsequent Rab7 delivery

would be lost. This is easy to test by cloning and expressing GFP-fusions of both Ccz1 and Mon1 and would be highly informative. Note we do not exclude canonical Rab7 recruitment in our model (see discussion), our data just indicate this has a minor contribution.

Reviewer 3:

2. The focus is on their manuscript is loading of Rab7 on phagosomes, but there's no indication about Rab7 activation (GTP-loading). Would the RILP-C33 probe work in *Dictyostelium*? If not possible, the activation state of Rab7 should still be discussed. Despite Rab7 on other organelles in PIKfyve-inhibited cells, is this active or not?

The GTP-loading status of Rab7 is a good question, although the general dogma is that membrane-localised Rabs are active. We will try the RILP-C33 probe in *Dictyostelium* as suggested, but as these cells lack an endogenous RILP orthologue there is a high chance it will not work. Sadly, reliable tools to assess active Rab status are a general limitation for the field, so if the RILP-C33 probe does not work we will add this caveat to the discussion.

3. The authors need to better address the confusing kinetics of early Rab7 recruitment, followed by SnxA (Fig. 4G, same for VatM - Fig. 4I) - which is counterintuitive if PIKfyve activity is required to recruit Rab7. How do the authors explain this? Are phagosomes prevented from acquiring Rab7 in PIKfyve deficient cells because of a defect on phagosomes or the endo-lysosomes loaded with Rab7 (but not active).

We believe this again relates to the over-simplification of our model. Our data indicate both PIKfyve dependent and independent Rab7 recruitment. In contrast to the abrupt recruitment of SnxA at ~120 seconds (Vines *et al.* JCB 2023), both Rab7 and VatM accumulate gradually over time starting from almost immediately following engulfment (Buckley *et al.* 2019, and Figure 2F). Our data indicate that the first stage of this is PIKfyve independent, and is responsible for ~10% of the total Rab7/V-ATPase accumulation by both the imaging in this paper, and Western blot for V-ATPase on purified phagosomes in Buckley *et al.* PLoS pathogens 2019. The arrival of some Rab7/V-ATPase prior to PI(3,5)P2 therefore supports our model where there are multiple sources of Rab7.

As the reviewer quite rightly points out, interpretation of the defects observed in the absence of PIKfyve becomes complex and we cannot completely differentiate between a defect on the phagosome, or the Rab7 compartments that fuse with them (or indeed both). In fact, we already note that small Rab7 compartments that we observe in wild-type cells are much more sparse in PIKfyve mutants. Therefore whilst the requirement for PI(3,5)P2 in the clustering and fusion of macropinosomes with phagosomes is clear, additional effects on the PI(3,5)P2-independent Rab7 compartments cannot be excluded.

The experiments above using the RILP-C33 active Rab7 biosensor as well as observation of the Mon1/Ccz complex should further clarify this, but we will also add further discussion of these points.

3. Description of the revisions that have already been incorporated in the transferred manuscript

Please insert a point-by-point reply describing the revisions that were already carried out and included in the transferred manuscript. If no revisions have been carried out yet, please leave this section empty.

Reviewer 1:

Minor comments.

1. It is unclear how the experiment in Figure 3G was conducted. If microscopic analysis was involved, the corresponding images should be included.

We apologise that we overlooked this and have now added a full description in the materials and

methods (P8 L16-21). Fluorescence measurements were performed using a plate reader, so there are no images.

2. Page 11-Line 2, the sentence "there was no obvious clustering around the nascent phagosome (Figure 2D)." It is Figure 2E, not Figure 2D.

Corrected.

3. There is an inconsistency regarding the description of fluorescent fusion proteins. For example, both GFP (RFP)-2xFyve and 2xFyve-GFP (RFP), as well as GFP-Rab5 and Rab5-GFP, were used. Typically, placing GFP (or RFP) before a gene suggests N-terminal tagging, while placing it after the gene implies C-terminal tagging. The authors should clarify the position of the fluorescent tag and ensure consistency in their descriptions.

We apologise for this oversight, and have been through and corrected all fusion protein references accordingly.

4. One of the videos was not referred in the manuscript or described in the Video legends. This video seems to correspond to Figure 5A, albeit with a different pseudo-color scheme.

This has been corrected. Video 7 does correspond to Fig 5A, and we have corrected the colour scheme to match and added references to the video in the text and figure legend.

Reviewer 2:

(2) In their abstract, the authors state that they "...delineate multiple subpopulations of Rab7-positive endosomes that fuse sequentially with phagosomes" (page 2, line 14,15). However, the data provides only evidence for V-ATPase or PI(3,5)P2-containing structures and the authors conclude to my understanding that macropinosomes are the main source for vesicular structures fusing with phagosomes. I would ask the authors to please be clear on the identity of the "Rab7-donor"-structures throughout the manuscript. Saying that they delineate multiple subpopulations of endosomes seems to be overstated.

We identify that macropinosomes are one source (subpopulation) of Rab7/PI(3,5)P2 vesicles but our data clearly show that they are the only source of Rab7 - there is clearly an additional early Rab positive / PI(3,5)P2-negative subpopulation of vesicles that cluster and fuse too at earlier stages. For example, in Figure 4F we co-express Rab7a/SnxA and show that whilst all the SnxA vesicles also contain Rab7 (and dextran), there is a clear separate population of small and early-fusing population of Rab7-containing vesicles that do not possess PI(3,5)P2. This is further validated in Figure 5B and C. To our mind this clearly demonstrates and defines different Rab7 endosomal populations, although we do not yet know the origins of the initial Rab7-positive/PI(3,5)P2 negative population - as discussed in our response to their point (3) below.

Minor points:

(1) The sentence "...which both deactivates and dissociates Rab5, and recruits and activates Rab7 on endosomes" is at least problematic as it suggests that Mon1-Ccz1 directly drives GTP-hydrolysis of Rab5 and dissociates it from the membrane. Indeed, Mon1-Ccz1 is shown to interfere with the positive feedback loop of the Rab5-GEF by interacting with Rabex (Poteryaev et al., 2010), so a rather indirect effect of Mon1-Ccz1. A GAP and the GDI are needed for Rab5 deactivation and dissociation from the membrane. How both are involved in the endosomal Rab-conversion is not clarified.

We have changed the text to better represent this complexity (P4 L4-6)

(2) Signals of RFP-labeled proteins are difficult to interpret throughout the experiments. What are the structures that show a strong accumulation of red signal in Fig. 1A,B, Fig 2G and Fig4A (20sec.) If these are fluorescently labeled proteins it would suggest that most of the proteins cluster/accumulate in the cell. Can the authors provide better images?

We appreciate that some of these reporters with multiple localisations can be difficult to interpret. This is major challenge for these sort of studies and main reason we use the large and easily-identified yeast containing phagosomes for quantification. In Fig. 1 the large structure is the large peri-nuclear cluster of Rab5 previously reported (Tu et al. JCB 2022). In Fig. 2G the bright structure is the recruitment of V-ATPase on the CV. Both these large structures easily distinguished from the phagosomal pool we are interested in. Whilst we would love to provide better images, this is simply not possible - both these other structures are unavoidable and we are already using some of the best microscopy methods available. We have however clarified the additional localisations seen in these images in the revised figure legends.

(3) On page 11 the authors state "...macropinosomes in Δ PIKfyve cells still appeared much larger. Quantification of their size and fluorescence intensity demonstrated that although macropinosomes started off the same size,...". This statement is not reflected in the data depicted in Fig. 3A,B. The size of the single labeled macropinosome appears to be larger in wildtype than in Δ PIKfyve cells from the beginning on. However, the quantification in Fig 3F is clear. So, are these bad examples in 3A,B, are they swapped or is this due to the additional expression of GFP-Rab7A? Could you please comment on the effect that the (over-)expression of GFP-tagged Rab-GTPases might have on the observations described in this paper in the discussion part?

As you can see from the error bars in Figure 3F, macropinosomes are extremely variable in size - ranging from ~0.2-5 microns in size in axenic *Dicytostelium*. The image in Figure 3B is therefore indicative of this heterogeneity, rather than being a "bad example". This is why we designed the experiment to quantify several hundred vesicles in order to make any conclusions - as well as doing it in the absence of any GFP- fusion expression.

Although we have not noticed any issues (enlarged vesicles are also clear in GFP-Rab7 expressing cells in Figure 1B), we do of course accept that GFP-Rab7 expression itself may have some detrimental effects on maturation and this is why we quantified macropinosome size in untransformed cells. We have clarified this in the results section (P12 L28).

(4) In Fig. 6E it is hard to distinguish if the dextran is accumulating inside the phagosome. I would suggest conducting a 3D reconstruction of these images to allow judging if macropinosomes fused with the phagosomes or if they cluster around the neck of the phagosome.

This would be nice, but not possible as these images are from single confocal sections, rather than a complete high-resolution Z-stack. We have however added an enlargement of both Figure 6D and E which we feel now more clearly shows the presence of dextran within the bounding PI(3)P membrane of the phagosome.

(5) In the discussion, the authors state that the small pool of "PIKfyve-independent Rab7" is "insufficient to for subsequent fusion with other Rab7A-positive compartments, further Rab7 enrichment, and lysosomal fusion." What is the rationale for this conclusion? Is it shown how many Rabs are necessary to induce a tethering and fusion event? It would be good to revise this part of the discussion also in respect of the first major point of my comments above.

Our show that in the absence of PIKfyve, phagosomes still remove Rab5 and gain a small pool of Rab7 but progress no further. This is consistent with some block in the HOPS-mediated homotypic fusion of Rab7 compartments. However, we accept that this is not necessarily due to simply not having enough Rab's so have rephrased the discussion accordingly.

(6) The intention of the paragraph about phagosomal ion channels is for this reviewer somehow out of context. It is not clear to me how the authors relate this to their findings. It would be could to bring this into a broader context.

We mention ion channels in the background as they represent the main class of PI(3,5)P2 effectors known so far. We feel this is important background context, even if our studies do not directly relate to this.

Reviewer 3:

4. Their disclosure and use of statistics is incomplete and/or inconsistent, and potentially wrong in some cases. For example, the authors disclose the number biological repeats in a few experiments (Fig. 3C, F) but not in the majority. Instead, they state the number of phagosomes without indicating biological repeats (eg. Fig. 2 and others). So, it is not possible to know if their data are reproducible. Despite not indicating independent experiments in some cases, they speak of SEM, which applies to mean of means from biological repeats. In other cases, none of this is disclosed (eg Fig. 3G). Often there is no indication of what statistical test was done OR if a statistical test was done (eg. Fig. 3G, Fig. 4, etc). I would recommend the authors review the excellent resource paper published in JCB on SuperPlots to better follow statistical expectations. This is essential to improve reproducibility and confidence in their observations.

We apologise if this was unclear for the referee, but we have tried to be clear in each case. The confusion likely lies in the definition of a biological repeat, which depends on the type of experiment. For quantification of phagocytic events over time, we feel it reasonable to take each individual event (each from an individual organism) as a biological repeat. This is because events are relatively rare and taken from multiple different movies, and it is not technically possible to film both mutants and controls simultaneously. In all these sort of experiments (e.g. Figure 2) we have shown standard deviation, which indicates the reproducibility between phagocytic events. We have clarified that these events are from movies obtained on at least 3 independent days in the methods.

In other cases, such as Figure 3C and F and Figures 5-6, we are able to take measurements across multiple cells simultaneously at each timepoint. It is therefore appropriate to average over multiple independent experimental repeats rather than individual cells. We have therefore used SEM in our analysis, and both the number of individual cells and independent repeats are stated on the graphs and legend. This was incomplete in a few cases but has now been clarified in all cases.

Regarding statistical tests, which ones were used now been clarified in each figure legend. Note that in Fig 3G, we do not apply any test as both lines essentially overlap and it is clear there would not be any convincing differences. In Figure 4, the graphs all compare co-expression of different reporters rather than different mutants or conditions and are from single events. We therefore feel statistical tests are unnecessary and inappropriate. Comparison of the same reporters between strains averaged across multiple events, with statistical analysis is shown in Fig 2 instead. All these points have now been added to the statistics section of the methods (P9 L1-6)

Minor Comments

6. It is interesting that 2FYVE-GFP stays on phagosomes for 50 min or more - this is distinct from macrophages. Please comment. Have the authors tried other PI(3)P probes to see if the same (PX-GFP).

We have not used other probes but we have no reason to believe 2xFYVE does not behave as predicted as it is the same probe used for most macrophage studies (FYVE domain from human Hrs), and gets removed from macropinosomes exactly as expected. We did not originally comment in this manuscript but PI3P dynamics are even more interesting as our previous data indicate that latex-bead containing phagosomes lose PI3P after 10 minutes (Buckley et al 2019, Figure 4F-G) This indicates phagosome maturation can be regulated by the cargo (under further investigation). Importantly however, both bead and yeast-containing phagosomes have comparable defects in the absence of PIKfyve. This is more fully discussed in our previous paper (Vines et al. JCB 2023) where we characterise PI(3)P and PI(3,5)P2 dynamics in more detail.

7. Fig. 7 model: the macropinosome in the diagram seems like a dead end as depicted - is there any arrow or change that could be added to show that it doesn't just sit there in the middle? Also, the light green on yellow hurts the eyes!

We apologise, there was actually supposed to be an arrow there but it was lost somewhere in the drafting process. The whole figure has now been updated to more clearly describe our full

and more complex model.

8. Fig. 3F, could be converted to volume assuming macropinosomes are spheres.

This is true, however as these images are taken from single planes we cannot know where in the sphere the slices are and therefore what the maximum diameter would be. We therefore prefer to keep it as area so as not to confuse and over-interpret the data.

9. Pg. 10, line 10 - Vps34 is Class III PI3K, not Class II.

Corrected.

4. Description of analyses that authors prefer not to carry out

Please include a point-by-point response explaining why some of the requested data or additional analyses might not be necessary or cannot be provided within the scope of a revision. This can be due to time or resource limitations or in case of disagreement about the necessity of such additional data given the scope of the study. Please leave empty if not applicable.

Reviewer 2:

(3) ("OPTIONAL") Optionally, the authors could also try to clarify these structures' identity by including further colocalization studies with additional early and late endosomal marker proteins. Are they for example positive for early or late endosomal markers like EEA1, ESCRT or Retromer? How about organelle-specific SNAREs? This would give further insights into the character of the "Rab7-donor" structures and would allow to clarify if multiple subpopulations are contributing to phagosome maturation in a sequential order as stated in the abstract. As I am not an expert on Dictyostellium I can't estimate the effort that would go into such an experimental setup. However, since the time scale of the events in the cell is nicely worked out in this study, these colocalization studies would not need to be conducted as live-cell microscopy experiments.

This is a sensible suggestion that would in theory help define these populations. However many of these markers are poorly defined with respect to phagosomes and/or Dictyostelium. *Dictyostelium* does not possess an EEA orthologue, but our data also indicate that these vesicles do not possess PI3P so cannot be canonical early endosomes. We have previously characterised WASH/retromer and whilst it is recruited to phagosomes at around the time of Rab5/7 transition Retromer appears to be recruited from the cytosol and drive recycling rather than being delivered on endosomes that fuse (see King et al. PNAS 2016). We have also previously looked at ESCRT (Lopez-Jimenez et al. PLoS Pathogens 2018) which also does not appear to have any recruitment to early phagosomes that would be consistent with a Rab7-sub-population. The SNAREs are yet to be studied in any detail, as they are often too divergent to assign a direct mammalian orthologue.

Therefore, whilst this is a sensible suggestion, and something we would like to follow up in the future, this is not straight-forward and we feel outside the scope of the current study. We have however included additional discussion of this in the revised manuscript (P20 L21-26).

Reviewer 3:

Major Comments:

1. Based on the current data, I am not entirely convinced that Rab7 is delivered mostly by fusion with other compartments. At least the data as provided cannot exclude other models. For example, Rab7-containing organelles that cluster with phagosomes may form contact sites that provide a local environment to load cytosolic Rab7. There's also a possibility that some of their Rab7 clusters are membrane sub-domains and not vesicles. Or perhaps, there is a first wave of cytosolic Rab7 recruitment, which then initiates fusion with Rab7 compartments, i.e., there is a two-phase Rab7 recruitment. While this last possibility is consistent with recruitment of Rab7 by fusion (the second phase), the authors present a model that is too simplistic and conclusive based on the data. The authors may be right, but they need to strengthen their evidence towards their claim. Maybe EM could help determine some of these issues. Perhaps better would be the use of FRAP, photo-

activation, or optogenetics of Rab7. For example, if Rab7 is acquired on phagosomes after photobleaching clusters of Rab7, this would suggest a cytosolic Rab7 contribution, and if not, this would support their model. I recognize that these experiments are not necessarily trivial, but either the authors augment their data (as suggested or with other approaches) or significantly pare down their conclusions.

We agree with the Referee that we cannot completely exclude other models, and as we talk about in the discussion, we do not wish to do so. We apologise if the role of fusion was over-stated but the model we propose is as the referee suggests: there is likely an early first wave of canonical Rab7 recruitment from the cytosol that is independent of PIKfyve before the majority of Rab7 is subsequently delivered by fusion in a PIKfyve-dependent manner. Our data indicate that the second wave is both quantitatively and functionally more significant (see functional data in Buckley *et al.* 2019).

We do however agree with the referee that we cannot formally exclude things such as contact-site mediated recruitment from the cytosol or sub-domains but not fusion however there is no data to support these either. In contrast, the hypothetical clustered Rab7 contacts/subdomains often (but not always) contain the transmembrane V-ATPase complex (Figure 2G) which must be delivered by fusion.

However we do not wish to over-simplify our conclusions and as we state in the discussion, we do think there is probably a small amount of Rab7 recruited from the cytosol by the canonical pathway. We accept that our cartoon in Figure 7 is over-focussed on fusion so we have substantially revised this, as well as the discussion to give a more balanced and complex view.

Regarding the proposed experiments, unfortunately, the imaging required to acquire these movies is already at the very limit of what is possible so we do not believe it would be technically feasible to employ methods such as FRAP and optogenetics on these relatively fast-moving phagosomes with the temporal resolution required. Furthermore, to differentiate recruitment from a cytosolic pool, every GFP-Rab7 cluster would need to be photobleached, which could not be reliably achieved.

However, this point will be largely addressed by the suggestion of Reviewer 2 to look at the Mon1/Ccz complex. The presence or absence of this will give strong evidence for canonical Rab5/7 transition and Rab7 recruitment from the cytosol which would significantly clarify our model and define the two different mechanisms of Rab7 recruitment to phagosomes.

5. Early macropinosomes fuse with early phagosomes more readily than 10-min old macropinosomes. Do 10-min old macropinosomes not fuse with older phagosomes? Is this not an issue of mismatched age?

This is an interesting point that we have clarified in the text. We agree with reviewer that it appears the ages of the macropinosomes and phagosomes must match but our data indicate this only occurs when both parties possess PI(3,5)P₂ as macropinosome fusions appears to happen in a single burst at about 240 seconds (Figure 6F) rather than as a continuous process. We also do not start to see any fusion of these older macropinosomes when the phagosomes get past the initial first 10 minutes of maturation (Figure 6G).

Original submission

First decision letter

MS ID#: JOCES/2023/261824

MS TITLE: PIKfyve is required for phagosomal Rab7 acquisition and the delivery and fusion of early macropinosomes to phagosomes

© 2026. Published by The Company of Biologists under the terms of the Creative Commons Attribution License (<https://creativecommons.org/licenses/by/4.0/>).

AUTHORS: James H Vines, Catherine M Buckley, and Jason Stewart King

ARTICLE TYPE: Research Article

Dear Jason,

We have now reached a decision on the above manuscript that you transferred to us from Review Commons.

I can see that you have already made substantial changes to the manuscript and consider your revision plan to be sensible. If you can provide further support for your conclusion in this was then I would be pleased to see a revised manuscript. We would then return it to the reviewers.

I have based this decision on the transferred reviews and your response so there are no new reviewer comments at this stage.

First revision

Author response to reviewers' comments

Vines et al. response to reviewers comments:

We thank the reviewers for their time and both thoughtful and constructive comments. We also apologise for the delay in completing this revision. Their specific points are addressed below but a general point that we would like to comment on is that in the original version it appears we did not make our model clear enough. The dogma in the field is that Rab7 is recruited to endosomes from a cytosolic pool via exchange with Rab5 (mediated by Mon1/Ccz1). Our work instead indicates that a significant proportion of Rab7 is also delivered to *Dictyostelium* phagosomes by fusion with other endocytic compartments, in a PIKfyve-dependent manner. It was not our intention to imply there was no canonical recruitment of Rab7 from an alternative cytosolic pool, and indeed we provide data to show this also happens and discuss this in the manuscript. Nonetheless, we accept we overstated the exclusivity of PIKfyve-dependent Rab7 recruitment at several points (including the abstract) and in our original model cartoon, and have tried to better explain a more nuanced model with multiple routes for Rab7 acquisition in this revision, including a completely redrawn model figure (new Fig. 7). Nonetheless, our demonstration that PIKfyve activity is required for fusion of nascent phagosomes with other compartments such as macropinosomes holds true with important functional consequences for phagosome maturation and microbial killing.

We respond to specific points below, author responses in blue:

Reviewer #1 (Evidence, reproducibility and clarity (Required)):

PIKfyve/Fab1, a kinase responsible for phosphorylating PI3P to produce PI(3,5)P2, regulates phagosome maturation across various organisms. A previous work from the authors' group demonstrated that disrupting PIKfyve in *Dictyostelium* inhibits the delivery of V-ATPase and hydrolase, thus dramatically reducing the ability of cells to acidify newly formed phagosomes and digest their contents. The current manuscript further dissects the function of PIKfyve and PI(3,5)P2. Using live cell imaging, the authors show that nascent phagosomes acquire Rab7 by fusion with multiple populations of Rab7-positive vesicles, and the loss of PIKfyve abolishes this event. One of these fusogenic vesicle populations was identified as PI(3,5)P2-positive macropinosomes, which dock and fuse with phagosomes in a PIKfyve-dependent manner. Based on these findings, the authors propose that PIKfyve contributes to phagosome maturation by promoting heterotypic fusion between phagosomes and macropinosomes, which help deliver regulatory components necessary for phagosome acidification and digestion. This study provides fresh insights into the process of

phagosome maturation. The work was well designed, performed and presented, and the manuscript is clearly written. However, there are several questions that should be addressed to strengthen the conclusions of the manuscript.

Major points:

1. The observation that macropinosomes undergo retrograde fusion with newly formed phagosomes to facilitate phagosome maturation is an interesting notion that challenges the traditional model. However, not all phagocytes exhibit a high level of macropinocytosis, and axenic *Dictyostelium* cells used in the study may be an exception. Thus, it remains unclear whether fusion with macropinosomes is universally required for phagosome maturation. WT *Dictyostelium* cells or axenic cells cultured under SorMC/Ka condition (Paschke et al., PLoS One, 2018) exhibit significantly reduced macropinocytosis. The authors could examine whether the accumulation of Rab7 and V-ATPase on large-sized phagosomes is delayed in these cells. These experiments may help broaden the applicability of the authors' finding.

This is a good suggestion, and we have now performed experiments to address this. Unfortunately, non-axenic *Dictyostelium* strains are unable to efficiently phagocytose large particles such as yeast (Bloomfield et al. eLife 2015), and we found that even bacterially grown axenic strains do this poorly - especially PIKfyve mutants. We were therefore unable to directly measure Rab5/V-ATPase recruitment under those conditions. However, instead we compared the proteolytic activity of cells grown on bacteria compared to axenic growth (new Figure S4C and D, p17 L20-30). As expected, macropinocytosis is reduced by >80% in both wild-type and PIKfyve- cells when grown on bacteria. However, phagosomal proteolysis was still almost non-existent in bacterially grown PIKfyve mutants. This is consistent with our previous work where we show that PIKfyve mutants also grow very poorly on bacteria (Buckley et al. PloS Pathogens 2019).

This demonstrates that even cells performing low levels of macropinocytosis require PIKfyve for phagosome maturation and further underlines that PIKfyve activity drives fusion with multiple vesicle classes, of which macropinosomes are only one. We have also clarified this in the text, both throughout the results, and in the revised discussion (e.g. P19, L11-14).

2. PIKfyve seems to play a specific role in the maturation of phagosomes but not macropinosomes. The differences may be driven by signaling from phagocytic receptors, as the author suggested. Alternatively, the large size of the yeast-containing phagosomes may require additional steps for efficient lysosomal delivery. The authors should consider examining whether PIKfyve is needed for the delivery of Rab7 and V-ATPase to phagosomes of comparable size to regular macropinosomes, such as those containing *K. aerogenes* or small beads. In addition, whether the process also involves fusion between phagosomes and macropinosomes should be verified.

This is again a good point and as suggested we have examined Rab7 recruitment to phagosomes containing bacteria (new Figure S2, P11 L30-34. We used *E. coli* rather than *K. aerogenes* as they have a distinctive rod shape that is easier to identify in images. Though we see significant recruitment of Rab7 to bacteria-containing phagosomes at later timepoints, its appearance is significantly delayed, consistent with our previous data where we show the killing of these bacteria is also delayed by a similar degree, and our measurements of phagosomal proteolysis, which were obtained using 3 μ m DQ-BSA beads (Buckley et al. PloS pathogens 2019). PIKfyve is therefore important for maturation of phagosomes of all sizes and cargos.

There is however clearly more GFP-Rab7A visible on bacterial phagosomes than we observe with yeast. Unfortunately, the small size of bacteria-containing phagosomes and asynchronous uptake prevents us from performing the same detailed quantification of Rab7 intensity over time, but this is consistent with the presence of both PIKfyve-dependent and independent pools of Rab7, where the PIKfyve-independent pool is limited and is spread more thinly over the larger membrane of a yeast-containing phagosome. (P11 L30, and Discussion P19 L16-22)

We also address whether bacteria-containing phagosomes still fuse with macropinosomes, by adding RFP-*E. coli* to cells after a 5-minute pulse of far-red dextran (new Figure S4A and B, P16 L27-32). From 5-minutes after addition of bacteria we could clearly see phagosomes containing both

bacteria and dextran, with the dextran occupying the space between the bacteria and the bounding phagosomal membrane (indicated by GFP-2xFYVE). This is a technically challenging experiment, and was not observed in all phagosomes due to incomplete and asynchronous macropinosome labelling combined with asynchronous bacterial uptake, so it is not possible to reliably quantify the frequency of this occurrence. Nonetheless we never observed this in 3 experiments with PIKfyve mutants, and it clearly shows that bacteria-containing phagosomes also fuse with macropinosomes in wild-type cells and it is not specific to large yeast-containing phagosomes.

3. In the previous study from the authors' group (Buckley et al., PLoS Pathog, 2019), it was shown that the accumulation of V-ATPase on phagosomes begins immediately after internalization in both PIKfyve mutant and WT, although V-ATPase accumulation reaches only half of the levels seen in WT. This partial accumulation of V-ATPase differs from the almost complete absence of Rab7 recruitment found in this study, which raises the question of whether there exists yet another population of fusogenic vesicles that are positive for V-ATPase but negative for Rab7. This could be checked by simultaneously examining the dynamics of V-ATPase and Rab7 during yeast phagocytosis in the PIKfyve mutant.

We agree with the referee that there are multiple pools of V-ATPase, and we show that there is both a very early PIKfyve-independent recruitment of both V-ATPase and Rab7 as well as a later and more substantial pool delivered in a PIKfyve-dependent manner. It is clear that V-ATPase and Rab7 do not always co-localise however - the clearest example being on the contractile vacuole, which has lots of V-ATPase but no Rab7 (the large bright magenta structure in Fig 2G.).

We suspect that the dramatically reduced, but not completely absent levels of both V-ATPase and Rab7 recruitment in the absence of PIKfyve are similar (in the discussion we estimate about 10% of GFP-Rab7A remains on phagosomes in PIKfyve KO's), but the challenges with imaging these very low levels means we cannot formally exclude that there is a pool of V-ATPase vesicles that lack Rab7 which fuse to very early phagosomes. Unfortunately, although we are able to co-express Rab7 and V-ATPase reporters well in Ax2 cells, we were unable to select any transformants with the same plasmid in PIKfyve mutants. This likely indicates that the reporters themselves are slightly dominant negative, and whilst wild-type cells can cope, this becomes lethal in the already-perturbed PIKfyve mutants. We therefore now acknowledge the possibility of an additional Rab7-independent V-ATPase pool in the discussion (p19 L27-34)

Minor points:

1. It is unclear how the experiment in Figure 3G was conducted. If microscopic analysis was involved, the corresponding images should be included.

The fluid-phase proteolysis was measured using a fluorescence plate-reader, following a 2-minute pulse of DQ-BSA, so there are no images. This has been clarified in both the methods and figure legend (p8 L28-31).

2. Page 11-Line 2, the sentence "there was no obvious clustering around the nascent phagosome (Figure 2D)." It is Figure 2E, not Figure 2D.

Thanks, corrected.

3. There is an inconsistency regarding the description of fluorescent fusion proteins. For example, both GFP (RFP)-2xFyve and 2xFyve-GFP (RFP), as well as GFP-Rab5 and Rab5-GFP, were used. Typically, placing GFP (or RFP) before a gene suggests N-terminal tagging, while placing it after the gene implies C-terminal tagging. The authors should clarify the position of the fluorescent tag and ensure consistency in their descriptions.

This has now been updated and corrected throughout.

4. One of the videos was not referred in the manuscript or described in the Video legends. This video seems to correspond to Figure 5A, albeit with a different pseudo-color scheme.

This has been corrected. Video 7 does correspond to Fig 5A, and we have corrected the colour scheme to match and added references to the video in the text (p15 L15) and figure legend.

Reviewer #1 (Significance (Required)):

Disruption of PIKfyve results in severe defects in phagosomal maturation across different organisms, but the underlying mechanism remains unclear. This study demonstrates that PIKfyve plays a specific role in phagosome maturation by promoting heterotypic fusion between macropinosomes and newly formed phagosomes. These fusion events provide a means for the rapid delivery of lysosomal components to early phagosomes. The study challenges the conventional model of phagosome maturation and provides novel insights into the complex dynamics involved. Nonetheless, further investigations are needed to elucidate the exact role of PIKfyve/PI(3,5)P₂ in regulating vesicle fusion and to explore whether the proposed model can be applied to other endocytic pathways or cell types.

Reviewer #2 (Evidence, reproducibility and clarity (Required)):

Summary

Vines et al. investigate in their manuscript the contribution of the lipid kinase PIKfyve on the maturation of phagosome. They follow the fate of the two main classes of identity markers of endocytic organelles: PIPs (PI(3)P and PI(3,5)P₂) and Rab-GTPases (Rab5 and Rab7) in wildtype and Δ PIKfyve Dictyostelium discoideum cells. To follow the two species of PIPs they utilize the established reporter-proteins GFP-2xFYVE (PI(3)P-binder) and GFP-SnxA (PI(3,5)P₂-binder) and correlate them with the appearance of fluorescently tagged Rab-GTPases on membranes of phagosomes in live-cell imaging.

They find that the deletion of PIKfyve does not alter the recruitment and behaviour of Rab5. Therefore a lack of PI(3,5)P₂ does not affect the early stages of phagosome formation. However, later stages of phagosome maturation are apparently affected by the lack of PI(3,5)P₂: the Rab-GTPase Rab7 is not localizing to the membrane of the phagosome. Closer inspection of their live cell imaging data led the authors to the conclusion that Rab7 is delivered by the fusion of Rab7-positive structures with the phagosomes in wildtype cells. This fusion seems to be dependent on PIKfyve and PI(3,5)P₂. However, surprisingly, the recruitment of Rab7 to macropinosomes or endosomal structures is independent of PIKfyve. The authors conclude (i) that lysosomal components are delivered to phagosomes by fusion of PI(3,5)P₂-positive macropinosomes and (ii) a non-canonical delivery from Rab7 to phagosomes by fusion instead of GEF-dependent recruitment from the cytosol.

Major comments

Overall, the submitted manuscript of Vines et al. is of very good quality. The presented data supports mainly the conclusions that the authors draw. Methods and statistical analysis are sound and well-described. Their rationale, the description of results, and the presentation of data are easy to follow and understand. However, there are two major points that I would like to address here:

(1) The authors show that deletion of PIKfyve results "in an almost complete block in Rab7 delivery to phagosomes" (page 17) indicating that the delivery of Rab7 depends on fusion with Rab7-positive structures. This would suggest that the Rab7-GEF Mon1-Ccz1 is not localized to the membrane of the phagosomes. Could the authors test for the presence of Mon1-Ccz1 in either fluorescence microscopy experiments or on purified phagosomes to exclude the possibility of a "canonical" Rab7 recruitment by its GEF? If the GEF is found on phagosomal membranes it would indicate that a Rab-transition from Rab5 to Rab7 occurs on the phagosome during maturation, but on a low level. The later fusion event might be a homotypic fusion of two Rab7-positive compartments. The observed fusion events could still deliver the bulk of Rab7 and other endolysosomal proteins to the phagosome. If the Rab7-GEF is not found on phagosomes how do the authors envision that the organelle keeps its identity? Is it solely dependent on PI(3,5)P₂? What is the fate of the Rab7-

negative phagosome in Δ PIKfyve cells if Rab7 is not delivered to the membrane, is there degradation happening over longer periods of time?

These are all good points for the reviewer and we thank them for the constructive suggestions. It seems we did not make it clear enough that our model is that there are multiple sources of Rab7, with the residual PIKfyve-independent Rab7-accumulation we report likely due to canonical Mon1/Ccz1-mediated exchange as we mentioned in the discussion but omitted from the model diagram.

Mon1 and Ccz1 are highly conserved, with clear *Dictyostelium* orthologues that have never been studied. However despite extensive efforts, we were unable to observe localisation of either protein, despite expressing multiple fusions and tagging at either end. This may be indicative of a low level of recruitment, masked by the cytoplasmic pool or some other technical reason but we are unfortunately unable to conclude anything about the presence of this canonical pathway, or its dependence on PIKfyve. We have therefore added this caveat to the discussion (p19 L22-27) and updated our model diagram accordingly (updated Figure 7).

(2) In their abstract, the authors state that they "...delineate multiple subpopulations of Rab7-positive endosomes that fuse sequentially with phagosomes" (page 2, line 14,15). However, the data provides only evidence for V-ATPase or PI(3,5)P2-containing structures and the authors conclude to my understanding that macropinosomes are the main source for vesicular structures fusing with phagosomes. I would ask the authors to please be clear on the identity of the "Rab7-donor"-structures throughout the manuscript. Saying that they delineate multiple subpopulations of endosomes seems to be overstated.

We identify that macropinosomes are one source (subpopulation) of Rab7/PI(3,5)P2 vesicles but our data clearly show that they are not the only source of Rab7 - there is clearly an additional early Rab7 positive / PI(3,5)P2-negative subpopulation of vesicles that cluster and fuse at earlier stages, and we still see comparable phagosomal proteolysis defect in cells with significantly reduced macropinocytosis (new Figure S4). For example, in Figure 4F we co-express Rab7a/SnxA and show that whilst all the SnxA vesicles also contain Rab7 (and dextran), there is a clear separate population of small and early-fusing population of Rab7-containing vesicles that do not possess PI(3,5)P2. This is further validated in Figure 5B and C. To our minds this clearly demonstrates and defines different Rab7 endosomal populations, although we do not yet know the origins of the initial Rab7-positive/PI(3,5)P2 negative population (discussed P19 L11-15, P19 L27-34).

("OPTIONAL") Optionally, the authors could also try to clarify these structures' identity by including further colocalization studies with additional early and late endosomal marker proteins. Are they for example positive for early or late endosomal markers like EEA1, ESCRT or Retromer? How about organelle-specific SNAREs? This would give further insights into the character of the "Rab7-donor" structures and would allow to clarify if multiple subpopulations are contributing to phagosome maturation in a sequential order as stated in the abstract. As I am not an expert on *Dictyostelium* I can't estimate the effort that would go into such an experimental setup. However, since the time scale of the events in the cell is nicely worked out in this study, these colocalization studies would not need to be conducted as live-cell microscopy experiments.

This is a sensible suggestion that would in theory help define these populations. However many of these markers are poorly defined with respect to phagosomes and/or *Dictyostelium*. *Dictyostelium* does not possess an EEA orthologue, but our data also indicate that these vesicles do not possess PI3P so cannot be canonical early endosomes. We have previously characterised WASH/retromer and whilst it is recruited to phagosomes at around the time of Rab5/7 transition Retromer appears to be recruited from the cytosol and drive recycling rather than being delivered on endosomes that fuse (see Buckley et al. PNAS 2016). We have also previously looked at ESCRT (Lopez-Jimenez et al. PLoS Pathogens 2018) which also does not appear to have any recruitment to early phagosomes that would be consistent with a Rab7-sub-population. The SNAREs are yet to be studied in any detail, as they are often too divergent to assign a direct mammalian orthologue.

Therefore, whilst this is a sensible suggestion, and something we would like to follow up in the future, this is not straight-forward and we feel outside the scope of the current study. We have however included additional discussion of this in the revised manuscript (P19 L27-34).

Minor comments

Minor points:

(1) The sentence "...which both deactivates and dissociates Rab5, and recruits and activates Rab7 on endosomes" is at least problematic as it suggests that Mon1-Ccz1 directly drives GTP-hydrolysis of Rab5 and dissociates it from the membrane. Indeed, Mon1-Ccz1 is shown to interfere with the positive feedback loop of the Rab5-GEF by interacting with Rabex (Poteryaev et al., 2010), so a rather indirect effect of Mon1-Ccz1. A GAP and the GDI are needed for Rab5 deactivation and dissociation from the membrane. How both are involved in the endosomal Rab-conversion is not clarified.

We have changed the text to better represent this complexity (P4 L4-6)

(2) Signals of RFP-labeled proteins are difficult to interpret throughout the experiments. What are the structures that show a strong accumulation of red signal in Fig. 1A,B, Fig 2G and Fig4A (20sec.) If these are fluorescently labeled proteins it would suggest that most of the proteins cluster/accumulate in the cell. Can the authors provide better images?

We appreciate that some of these reporters with multiple localisations can be difficult to interpret. This is major challenge for these sort of studies and main reason we use the large and easily-identified yeast containing phagosomes for quantification. In Fig. 1 the large structure is the large peri-nuclear cluster of Rab5 previously reported (Tu et al. JCB 2022). In Fig. 2G the bright structure is the recruitment of V-ATPase on the CV. Both these large structures are easily distinguished from the phagosomal pool we are interested in. Whilst we would love to provide better images, this is simply not possible - both these other structures are unavoidable and we are already using some of the best microscopy methods available. We have however clarified the additional localisations seen in these images in the revised figure legends.

(3) On page 11 the authors state "...macropinosomes in Δ PIKfyve cells still appeared much larger. Quantification of their size and fluorescence intensity demonstrated that although macropinosomes started off the same size,...". This statement is not reflected in the data depicted in Fig. 3A,B. The size of the single labeled macropinosome appears to be larger in wildtype than in Δ PIKfyve cells from the beginning on. However, the quantification in Fig 3F is clear. So, are these bad examples in 3A,B, are they swapped or is this due to the additional expression of GFP-Rab7A? Could you please comment on the effect that the (over-)expression of GFP-tagged Rab-GTPases might have on the observations described in this paper in the discussion part?

As you can see from the error bars in Figure 3F, macropinosomes are extremely variable in size - ranging from ~0.2-5 microns in size in axenic *Dicytostelium*. The image in Figure 3B is therefore indicative of this heterogeneity, rather than being a "bad example". This is why we designed the experiment to quantify several hundred vesicles in order to make any conclusions - as well as doing it in the absence of any GFP-fusion expression.

Although we have not noticed any issues (enlarged vesicles are also clear in GFP-Rab7 expressing cells in Figure 1B), we do of course accept that GFP-Rab7 expression itself may have some detrimental effects on maturation and this is why we quantified macropinosome size in untransformed cells. We have clarified this in the results section (P13 L7-9).

(4) In Fig. 6E it is hard to distinguish if the dextran is accumulating inside the phagosome. I would suggest conducting a 3D reconstruction of these images to allow judging if macropinosomes fused with the phagosomes or if they cluster around the neck of the phagosome.

This would be nice, but not possible as these images are from single confocal sections, rather than a complete high-resolution Z-stack. We have however added an enlargement of both Figure 6D and E which we feel now more clearly shows the presence of dextran within the bounding PI(3)P membrane of the phagosome.

(5) In the discussion, the authors state that the small pool of "PIKfyve-independent Rab7" is "insufficient to for subsequent fusion with other Rab7A-positive compartments, further Rab7 enrichment, and lysosomal fusion." What is the rationale for this conclusion? Is it shown how many Rabs are necessary to induce a tethering and fusion event? It would be good to revise this part of the discussion also in respect of the first major point of my comments above.

Our show that in the absence of PIKfyve, phagosomes still remove Rab5 and gain a small pool of Rab7 but progress no further. This is consistent with some block in the HOPS-mediated homotypic fusion of Rab7 compartments. However, we accept that this is not necessarily due to simply not having enough Rab's so have removed this statement from the discussion.

(6) The intention of the paragraph about phagosomal ion channels is for this reviewer somehow out of context. It is not clear to me how the authors relate this to their findings. It would be could to bring this into a broader context.

We mention ion channels in the background as they represent the main class of PI(3,5)P2 effectors known so far. We feel this is important background context, even if our studies do not directly relate to this, although we have rephrased slightly to try and emphasise the context better (P20 L18-26).

****Referees cross-commenting****

Reviewer #1 provides valid questions. Addressing them would improve the manuscript by allowing consideration if the findings only apply to Dictyostellium or is of broader interest.

I completely agree with the concern of Reviewer #3 that the data provided so far would also allow for alternative models. The authors need to include further controls to exclude Rab7 recruitment or activation by any other means than fusion.

Reviewer #2 (Significance (Required)):

2. Significance

The manuscript by Vines et al. describes a very interesting novel observation on how the organelle identity marker Rab7 is delivered to phagosomes. They propose a mechanism, the delivery of Rab7 by PIKfyve-dependent fusion events with Rab7-positive macropinosomes, which is in contrast to the canonical model that endosomal organelles gain their Rab7-identity by maturation from a Rab5-positive compartment with the help of the Rab7-GEF Mon1-Ccz1. In the proposed mechanism the lipid-kinase PIKfyve, which is also involved in cellular signaling processes, plays the key role. In this study the authors present profound live cell imaging experiments combined with pulse-chase uptake of phagosomal cargoes. The obtained data is giving surprising new insights on the order of events in the maturation of phagosomes and suggests an unprecedentedly important role for PIKfyve in the maturation process. These new insights are of broad interest to a readership interested in transport, maturation and signaling processes along the endolysosomal system as well as of interest in the perspective of pathogen invasion to host cells.

Reviewer #3 (Evidence, reproducibility and clarity (Required)):

In Vines et al., the authors used time-lapse imaging of Dictyostelium to investigate the spatial-temporal maturation of macropinosomes labelled with a short pulse of dextran and phagosomes using yeast particles. The phagocytes expressed fluorescent Rab5 and/or Rab7 and/or biosensors for PI(3)P using 2FYVE-GFP and PI(3,5)P2 using the authors recently disclosed SnxA. They quantified the dynamics of these probes in wild-type and PIKfyve-deleted cells. The authors provide evidence for their main observations, which are that: i) Rab5 and PI(3)P are acquired early and independently of PIKfyve on phagosomes and macropinosomes, ii) but phagosomes require PIKfyve to acquire Rab7, iii) that phagosomes acquire Rab7 by fusing with Rab7-containing vesicles that cluster around the phagosome, iv) that macropinosomes do not require PIKfyve for Rab7 acquisition, and v) that PI(3,5)P2 on phagosomes follows Rab7. While the imaging data is high quality and supports several of the claims, the major discovery as proposed here is not fully

supported by the data provided. I think the authors must address the following to strengthen their otherwise beautiful work.

Major Comments:

Major comments

1. Based on the current data, I am not entirely convinced that Rab7 is delivered mostly by fusion with other compartments. At least the data as provided cannot exclude other models. For example, Rab7-containing organelles that cluster with phagosomes may form contact sites that provide a local environment to load cytosolic Rab7. There's also a possibility that some of their Rab7 clusters are membrane sub-domains and not vesicles. Or perhaps, there is a first wave of cytosolic Rab7 recruitment, which then initiates fusion with Rab7 compartments, i.e., there is a two-phase Rab7 recruitment. While this last possibility is consistent with recruitment of Rab7 by fusion (the second phase), the authors present a model that is too simplistic and conclusive based on the data. The authors may be right, but they need to strengthen their evidence towards their claim. Maybe EM could help determine some of these issues. Perhaps better would be the use of FRAP, photo-activation, or optogenetics of Rab7. For example, if Rab7 is acquired on phagosomes after photobleaching clusters of Rab7, this would suggest a cytosolic Rab7 contribution, and if not, this would support their model. I recognize that these experiments are not necessarily trivial, but either the authors augment their data (as suggested or with other approaches) or significantly pare down their conclusions.

We agree with the reviewer that the model whereby Rab7 is solely delivered by fusion is over-simplistic and we apologise if we presented our model too strongly in this way. We always agreed with the reviewer that there are multiple sources of Rab7, and our data show that about 10% of the GFP-Rab7 signal on large, yeast containing phagosomes still accumulates independently of PIKfyve, and this proportion appears higher with smaller cargo such as bacteria. Nonetheless, PIKfyve is clearly important for fusion with Rab7-containing compartments, including (but not exclusively) macropinosomes.

Please see our responses to similar comments to the other reviewers, as well as the revised discussion and model figure (Figure 7, P19 L16-34). The suggested experiments would be nice but FRAP in particular would not be technically feasible given the number of vesicles (in 3D), their dynamism and the light-sensitivity of the amoebae. Nonetheless we hope that the clarification of the non-exclusivity of our model is sufficient.

2. The focus in their manuscript is loading of Rab7 on phagosomes, but there's no indication about Rab7 activation (GTP-loading). Would the RILP-C33 probe work in *Dictyostelium*? If not possible, the activation state of Rab7 should still be discussed. Despite Rab7 on other organelles in PIKfyve-inhibited cells, is this active or not?

This is a nice idea, for which we thank the reviewer. We tried this, and expressed a RILP-C33 probe in *Dictyostelium* but could not see any recruitment to membranes at all. This is likely because *Dictyostelium* doesn't have a RILP orthologue of its own and the mammalian version does not bind sufficiently to *Dictyostelium* Rab7. Measuring the activation status of Rab GTPases is a general issue in the field and we have instead added this caveat in the discussion (P20 L14-16)

3. The authors need to better address the confusing kinetics of early Rab7 recruitment, followed by SnxA (Fig. 4G, same for VatM - Fig. 4I) - which is counterintuitive if PIKfyve activity is required to recruit Rab7. How do the authors explain this? Are phagosomes prevented from acquiring Rab7 in PIKfyve deficient cells because of a defect on phagosomes or the endo-lysosomes loaded with Rab7 (but not active).

This again relates to the over-simplification of our model. Our data indicate both PIKfyve dependent and independent Rab7 recruitment. In contrast to the abrupt recruitment of SnxA at ~120 seconds (Vines *et al.* JCB 2023), both Rab7 and VatM accumulate gradually over time starting from almost immediately following engulfment (Buckley *et al.* 2019, and Figure 2F). Our data indicate that the first stage of this is PIKfyve independent, and is responsible for ~10% of the total

Rab7/V-ATPase accumulation by both the imaging in this paper, and Western blot for V-ATPase on purified phagosomes in Buckley *et al.* PLoS pathogens 2019. The arrival of some Rab7/V-ATPase prior to PI(3,5)P2 therefore supports our model where there are multiple sources of Rab7. This is updated in our model figure (Figure 7) and in the discussion (e.g. P19 L16 onwards)

4. Their disclosure and use of statistics is incomplete and/or inconsistent, and potentially wrong in some cases. For example, the authors disclose the number biological repeats in a few experiments (Fig. 3C, F) but not in the majority. Instead, they state the number of phagosomes without indicating biological repeats (eg. Fig. 2 and others). So, it is not possible to know if their data are reproducible. Despite not indicating independent experiments in some cases, they speak of SEM, which applies to mean of means from biological repeats. In other cases, none of this is disclosed (eg. Fig. 3G). Often there is no indication of what statistical test was done OR if a statistical test was done (eg. Fig. 3G, Fig. 4, etc). I would recommend the authors review the excellent resource paper published in JCB on SuperPlots to better follow statistical expectations. This is essential to improve reproducibility and confidence in their observations.

We apologise if this was unclear for the referee, but we have updated the manuscript to try to be clear in each case. The confusion likely lies in the definition of a biological repeat, which depends on the type of experiment. For quantification of phagocytic events over time, we feel it reasonable to take each individual event (each from an individual organism) as a biological repeat. This is because events are relatively rare and taken from multiple different movies, where it is not technically possible to film both mutants and controls simultaneously. In all these sort of experiments (e.g. Figure 2) we have shown standard deviation, which indicates the reproducibility between phagocytic events. We have clarified that these events are from movies obtained on at least 3 independent days in the methods.

In other cases, such as Figure 3C and F and Figures 5-6, we are able to take measurements across multiple cells simultaneously at each timepoint. It is therefore appropriate to average over multiple independent experimental repeats rather than individual cells. We have therefore use SEM in these analyses, with the means from each repeat shown as separate points, and both the number of individual cells and independent repeats are stated on the graphs and legend. This was incomplete in a few cases but has now been clarified throughout.

Regarding statistical tests, which ones used have now been clarified in each figure legend. Note that in Fig 3G, we do not apply any test as both lines essentially overlap and it is clear there would not be any convincing differences. In Figure 4, the graphs all compare co-expression of different reporters rather than different mutants or conditions and are from single events. We therefore feel statistical tests are unnecessary and inappropriate. Rigorous comparison of the same reporters between strains averaged across multiple events, with statistical analysis is shown in Fig 2 instead. All these points have now been added to the statistics section of the methods (P9 L1-6).

5. Early macropinosomes fuse with early phagosomes more readily than 10-min old macropinosomes. Do 10-min old macropinosomes not fuse with older phagosomes? Is this not an issue of mismatched age?

This is an interesting point that we have clarified in the text. We agree with reviewer that it appears the ages of the macropinosomes and phagosomes must match but our data indicate this only occurs when both parties possess PI(3,5)P2 as macropinosome fusions appears to happen in a single burst at about 240 seconds (Figure 6F) rather than as a continuous process. We also do not start to see any fusion of these older macropinosomes when the phagosomes get past the initial first 10 minutes of maturation (Figure 6G).

Minor Comments

6. It is interesting that 2FYVE-GFP stays on phagosomes for 50 min or more - this is distinct from macrophages. Please comment. Have the authors tried other PI(3)P probes to see if the same (PX-GFP).

We have not used other probes but we have no reason to believe 2xFYVE does not behave as predicted as it is the same probe used for most macrophage studies (FYVE domain from human Hrs),

and gets removed from macropinosomes exactly as expected. We did not originally comment in this manuscript, but PI3P dynamics are even more interesting as our previous data indicate that latex-bead containing phagosomes lose PI3P after 10 minutes (Buckley et al 2019, Figure 4F-G) This indicates phagosome maturation can be regulated by the cargo (under further investigation). Importantly however, both bead and yeast-containing phagosomes have comparable defects in the absence of PIKfyve. This is more fully discussed in our previous paper (Vines et al. JCB 2023) where we characterise PI(3)P and PI(3,5)P₂ dynamics in more detail.

7. Fig. 7 model: the macropinosome in the diagram seems like a dead end as depicted - is there any arrow or change that could be added to show that it doesn't just sit there in the middle? Also, the light green on yellow hurts the eyes!

We apologise, there was actually supposed to be an arrow there but it was lost somewhere in the drafting process. The whole figure has now been updated and aesthetically improved to more clearly describe our full and more complex model.

8. Fig. 3F, could be converted to volume assuming macropinosomes are spheres.

This is true, however as these images are taken from single planes we cannot know where in the sphere the slices are and therefore what the maximum diameter would be. We therefore prefer to keep it as area so as not to confuse and over-interpret the data.

9. Pg. 10, line 10 - Vps34 is Class III PI3K, not Class II.

Corrected

Reviewer #3 (Significance (Required)):

Overall, the potential novelty of this work is the authors' proposal that phagosomes acquire Rab7 mostly by fusion with Rab7-labelled organelles rather than a cytosolic pool. This is distinct from existing models that assume phagosomes acquire Rab7 from a cytosolic pool that is loaded onto the membrane. They also suggest that PIKfyve plays a role in this process. However, as noted above, this claim needs to stronger data as the current data allows for other possible models, in my opinion.

This work is of relevance to cell biologists interested in membrane trafficking, phagocytosis, model organisms, and microscopy.

Second decision letter

MS ID#: jcs.264814R1

MS Title: PIKfyve is required for efficient phagosomal Rab7 acquisition and the delivery and fusion of early macropinosomes to phagosomes

Authors: James Vines; Catherine Buckley; Ilona Willson; Daniel Stark; Jason S. King

Article Type: Research Article

Dear Dr King,

I am happy to tell you that your manuscript has been accepted for publication in Journal of Cell Science, pending standard publication integrity checks.

Thank you for sending your manuscript to Journal of Cell Science through Review Commons.